# Chromatin-associated RNA sequencing (ChAR-seq) maps genome-wide RNA-to-DNA contacts

Jason C Bell[1][†]*, David Jukam[2], Nicole A Teran[1,3], Viviana I Risca[3], Owen K Smith[1,4], Whitney L Johnson[1], Jan M Skotheim[2], William James Greenleaf[3,5], Aaron F Straight[1,4]*

[1]Department of Biochemistry, Stanford University, Stanford, United States; [2]Department of Biology, Stanford University, Stanford, United States; [3]Department of Genetics, Stanford University, Stanford, United States; [4]Department of Chemical and Systems Biology, Stanford University, Stanford, United States; [5]Department of Applied Physics, Stanford University, Stanford, United States

**Abstract** RNA is a critical component of chromatin in eukaryotes, both as a product of transcription, and as an essential constituent of ribonucleoprotein complexes that regulate both local and global chromatin states. Here, we present a proximity ligation and sequencing method called **Ch**romatin-**A**ssociated **R**NA **seq**uencing (ChAR-seq) that maps all RNA-to-DNA contacts across the genome. Using *Drosophila* cells, we show that ChAR-seq provides unbiased, de novo identification of targets of chromatin-bound RNAs including nascent transcripts, chromosome-specific dosage compensation ncRNAs, and genome-wide trans-associated RNAs involved in co-transcriptional RNA processing.

DOI: https://doi.org/10.7554/eLife.27024.001

*For correspondence:
jason.bell@10xgenomics.com
(JCB);
astraigh@stanford.edu (AFS)

Present address: [†]Molecular Biology, 10x Genomics, Pleasanton, United States

## Introduction

Much of the eukaryotic genome is transcribed into non-coding RNA (ncRNA), and several studies have established that a subset of these ncRNAs form ribonucleoprotein complexes that bind and regulate chromatin (*Guttman and Rinn, 2012*; *Meller et al., 2015*; *Cech and Steitz, 2014*). Some of the most well studied ncRNAs are those involved in dosage compensation, which include *roX1* and *roX2* in *Drosophila* and *Xist* in mammals. In *Drosophila*, *roX1* and *roX2* are part of the male-specific lethal (MSL) complex that coats the single male X chromosome to acetylate histone H4K16 and increase transcription (*Conrad and Akhtar, 2012*). In female mammals, *Xist* is expressed from a single X-chromosomal locus and coats the X chromosome from which it is expressed in order to silence transcription (*Augui et al., 2011*). Other ncRNAs, such as *HOTAIR* (*Rinn et al., 2007*; *Chu et al., 2011*), *HOTTIP* (*Wang et al., 2011*), and enhancer RNAs (*Sigova et al., 2015*), have been shown to regulate expression of specific genes by localizing to chromatin and recruiting activating or repressing proteins. Finally, repetitive ncRNA transcripts have roles at chromosomal loci essential in maintaining genomic integrity over many cell divisions, including TERRA at telomeres (*Bunting et al., 2010*) and alpha-satellites near centromeres (*Hall et al., 2012*). Despite these well-studied examples, the genomic targets of most chromatin-associated ncRNAs are unknown, and the mechanisms by which these ncRNAs regulate the epigenetic and spatial organization of chromatin remain largely unexplored.

Genomic methods for studying the localization of specific RNA transcripts include ChIRP (*Chu et al., 2011*), CHART (*Engreitz et al., 2013*), and RAP (*Simon et al., 2011*). These techniques use hybridization of complementary oligonucleotides to pull down a single target RNA and then

next generation sequencing or mass spectrometry to identify its DNA- or protein-binding partners (*Simon et al., 2011*). However, de novo discovery of chromatin-associated RNAs remains limited to computational predictions (*Guttman and Rinn, 2012*) or association with previously known factors (*Khalil et al., 2009*). Nuclear fractionation allows isolation of bulk chromatin and subsequent identification of chromatin bound RNAs via sequencing, but does not provide sequence-resolved maps of RNA binding locations along the genome (*Werner and Ruthenburg, 2015*). To overcome these limitations, we have developed ChAR-seq, a proximity ligation and sequencing method (*Figure 1A*) that both identifies chromatin-associated RNAs and maps them to genomic loci (*Figure 1B*).

## Results

We developed and performed ChAR-seq using *Drosophila melanogaster* CME-W1-cl8+ cells, a male wing disc derived cell line with a normal karyotype and well-characterized epigenome and transcriptome (*Cherbas et al., 2011*; *Roy et al., 2010*). ChAR-seq is a chromosome conformation capture method that maps genome-wide RNA-to-DNA contacts in crosslinked nuclei (*Dekker et al., 2002*; *Rao et al., 2014*). Briefly, cells are cross-linked with formaldehyde and permeabilized, then RNA is partially fragmented and soluble RNA is washed away. The chromatin-cross-linked RNA is then ligated to an oligonucleotide duplex 'bridge' molecule and reverse transcribed. Genomic DNA is then digested and ligated onto the other end of the oligonucleotide 'bridge', creating a link between chromatin-associated RNA and proximal DNA. The ligated RNA is fully converted to cDNA during second strand synthesis. Finally, the DNA is sonicated and the chimeric molecules are purified, processed, and sequenced.

To enable the capture and analysis of RNA-to-DNA contacts, the oligonucleotide bridge (*see Figure 1—figure supplement 1*) was designed to have several key features: (1) the 5'-adenylated end (5'App) of the bridge ensures that it is the only 5' end competent for ligation to the 3'-ends of ssRNA by truncated T4 Rnl2tr R55K K227Q mutant ligase (*Viollet et al., 2011*), which cannot adenylate 5' ends (*Figure 1—figure supplement 2*), (2) the sequence of the bridge does not exist in the yeast, fly, mouse or human genomes and encodes a defined polarity, (3) the end of the bridge contains a restriction site for specific ligation to digested genomic DNA, and (4) the bridge is biotinylated so that it can be captured and enriched. After the bridge is ligated to RNA in situ, the molecules are stabilized by reverse transcription using Bst3.0 polymerase, which can traverse the DNA-RNA junction. The genomic DNA is then digested using the restriction enzyme DpnII, which has a median spacing of ~200 bp between sites in the fly genome. The digested genomic DNA is then ligated to the bridge adaptor using T4 DNA ligase. Second strand synthesis is completed using limiting RNase H and DNA Polymerase I. Crosslinks are then reversed followed by DNA precipitation, sonication and purification using streptavidin beads. Libraries are constructed by repairing and dA-tailing the DNA fragments, ligating TruSeq adaptors to the ends and PCR amplifying (*Figure 1A*).

Upon conversion of RNA-DNA contacts to a covalent chimera, the chimeric molecules were sequenced using 152 bp single-end reads. Sequencing across the bridge junction ensures identification of the RNA and DNA portions of the chimeric molecule by reading the polarity of the bridge (*Figure 1B*). The RNA/cDNA (*Figure 1B*, *red*) and the genomic DNA side (*Figure 1B*, *black*) of each read were computationally split and aligned to the transcriptome and genome. After post-processing for unique alignments, repeat removal, and removal of blacklisted regions, each RNA molecule was mapped to the genomic location to which it was ligated (*see Materials and methods and Figure 1—figure supplement 3*), resulting in 22.2 million high-confidence unique mapping events for ~16,800 RNA transcripts. All individual RNA-to-DNA contacts for a given transcript were then combined to produce a genome-wide association map for each individual transcript (*Figure 1C*). To ensure that ChAR-seq signal was not due to spurious bridge-to-DNA ligation, we performed a control experiment in which we added RNase A and RNase H to lysed cells before the RNA-to-bridge ligation. This RNase-treatment reduced the number of unique bridge molecules identified by six-fold, demonstrating that the vast majority of bridge ligation events are indeed RNA-dependent (*Figure 1—figure supplement 4*). The ChAR-seq protocol is highly reproducible, with the number normalized RNA-to-DNA contacts observed for each RNA showing high concordance between replicates (*Figure 1—figure supplement 5*). To estimate the specificity of our observed RNA-to-DNA ligation events and to ensure that the contacts that we observed were not due to diffusion of

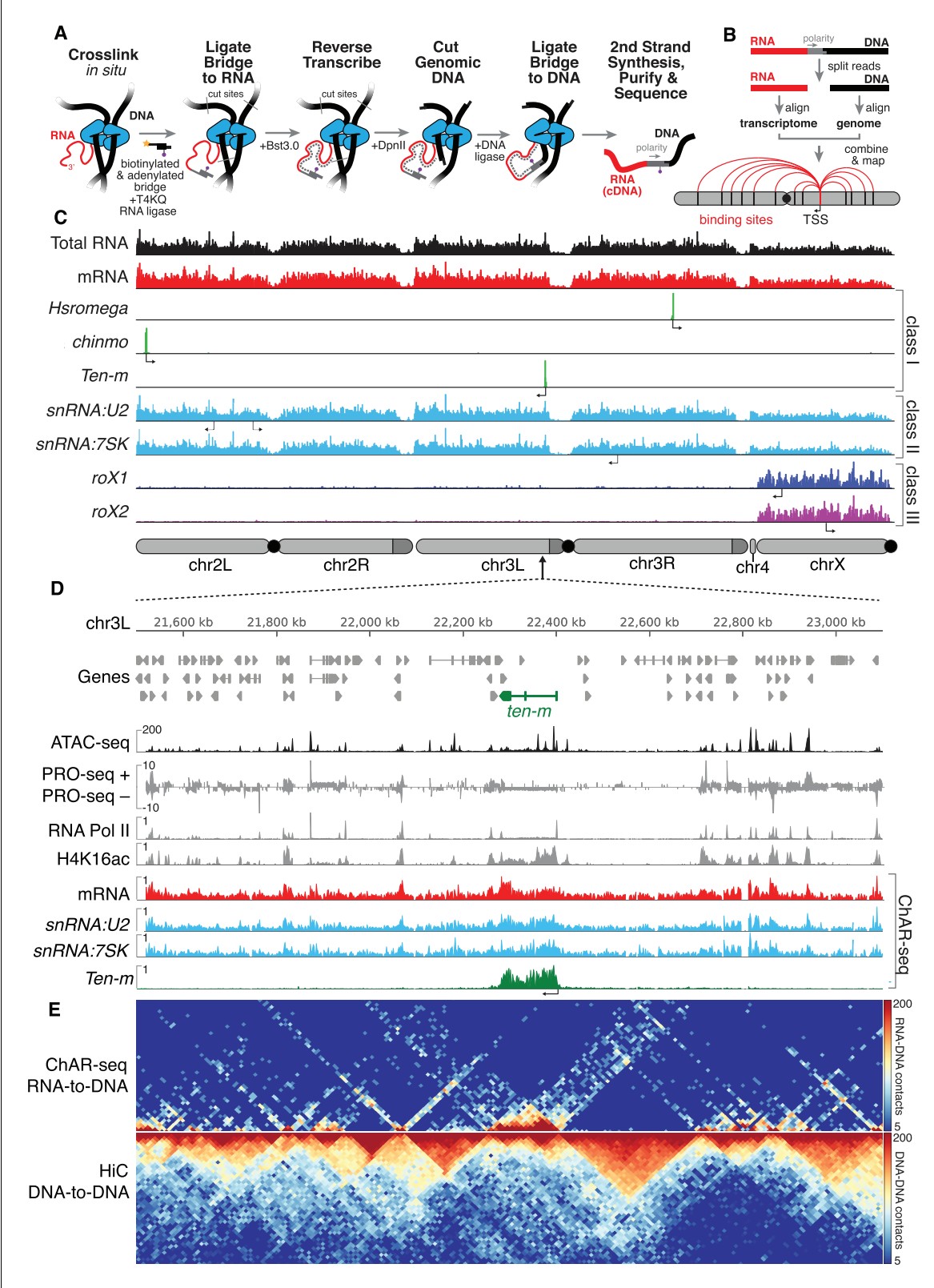

**Figure 1.** ChAR-seq uses proximity ligation of chromatin-associated RNA and deep sequencing to map RNA-DNA contacts in situ. (**A**) Overview of the ChAR-seq method wherein RNA-DNA contacts are preserved by crosslinking, followed by in situ ligation of the 3' end of RNAs to the adenylated 5' end of the ssDNA tail of an oligonucleotide 'bridge' containing a biotin modification and a DpnII-complementary overhang on the opposite end. After extending the bridge by reverse transcription to generate a strand of cDNA complementary to the RNA, the genomic DNA is then digested with DpnII

*Figure 1 continued on next page*

*Figure 1 continued*

and then re-ligated, capturing proximally-associated bridge molecules and RNA. The chimeric molecules are reverse-transcribed, purified and sequenced. (B) Chimeric molecules are sequenced and the RNA and DNA ends are distinguished owing to the polarity of the bridge, which preferentially ligates to RNA via the 5'-adenylated tail and to DNA via the DpnII overhang. The RNA and DNA reads are then computationally recombined to produce contact maps for each annotated RNA in the genome. (C) Representative examples of genome-wide RNA coverage plots generated for Total RNA (black), mRNA (red), *Hsromega* (green), *chinmo* (green), *ten-m* (green), *snRNA:U2* (cyan), *snRNA:7SK* (cyan), *rox1* (blue) and *roX2* (purple). Arrows show the transcription start site for each gene. In chromosome cartoons throughout the paper, light gray represents the primary chromosome scaffolds, darker gray regions are heterochromatic scaffolds, and black circles are centromeres. (D) Zoomed in region for an 850 kilobase region of chromosome 3L (chr3L). ChAR-seq tracks for Total RNA, *ten-m*, *snRNA:U2*, and *snRNA:7SK* are shown in comparison with PRO-seq tracks (*Drosophila* S2 [*Kwak et al., 2013*]) and ATAC-seq (this study, CME-W1-cl8+). (E) ChAR-seq contact matrix (RNA-to-RNA, *top*) plotted and aligned with same 850 kb region as panel D. ChAR-seq was performed without bridge addition (Hi-C/Mock-ChAR), resulting in DNA-DNA proximity ligation as in Hi-C ('Hi-C, DNA-to-DNA', *bottom*).

DOI: https://doi.org/10.7554/eLife.27024.002

The following figure supplements are available for figure 1:

**Figure supplement 1.** Diagram of the oligonucleotide bridge and efficiency of bridge ligation and capture.

DOI: https://doi.org/10.7554/eLife.27024.003

**Figure supplement 2.** In vitro optimization of RNA-to-DNA ligation conditions.

DOI: https://doi.org/10.7554/eLife.27024.004

**Figure supplement 3.** Diagram of the ChAR-seq data processing pipeline and bar plot of RNA alignment.

DOI: https://doi.org/10.7554/eLife.27024.005

**Figure supplement 4.** ChAR-seq RNA-to-bridge ligation is sensitive to RNase treatment.

DOI: https://doi.org/10.7554/eLife.27024.006

**Figure supplement 5.** Comparison of RNA-to-DNA contacts between replicates.

DOI: https://doi.org/10.7554/eLife.27024.007

**Figure supplement 6.** False positive contacts are proportional to RNA spike-in level.

DOI: https://doi.org/10.7554/eLife.27024.008

**Figure supplement 7.** Chromatin-associated RNA alignment by class.

DOI: https://doi.org/10.7554/eLife.27024.009

**Figure supplement 8.** Abundance of cis contacts.

DOI: https://doi.org/10.7554/eLife.27024.010

**Figure supplement 9.** ChAR-seq RNA-DNA contacts are dissimilar to DNA-DNA contacts.

DOI: https://doi.org/10.7554/eLife.27024.011

**Figure supplement 10.** ChAR-seq protocol preserves genome organization.

DOI: https://doi.org/10.7554/eLife.27024.012

RNA after fragmentation, we performed a spike-in experiment wherein we added increasing amounts of exogenous RNA to the cells after lysis but before bridge ligation. After lysis and RNA fragmentation, we recovered and quantified the total soluble RNA from the supernatant, then spiked-in purified in vitro transcribed RNA fragments (~200 nt) from commonly used protein expression vectors (MBP, HALO and GFP). The spike-in controls were added at 0.1%, 1% and 10% of the total, recoverable RNA by mass. Though we see a clear concentration-dependent increase in the number of false positives, even in the scenario where we spiked in RNA at 10% of the total soluble RNA we observed fewer than 0.5% false positive RNA-to-DNA contacts (*Figure 1—figure supplement 6*), which compares favorably to the false positive rates in related RNA-DNA mapping methods (*Li et al., 2017*; *Sridhar et al., 2017*).

Only the 3'-hydroxyl of each RNA is available for ligation to the bridge, thus the polarity of each RNA molecule with respect to its transcriptional direction can be determined by its orientation with respect to the bridge. The majority (85% of total) of the RNAs captured in our assay were sense, with the largest single subtype represented by sense-stranded mRNA (32% of total), owing to the capture of nascent transcripts (*Figure 1—figure supplement 7*). Most of the chromatin-associated antisense transcripts that we identified arose from ncRNA or intronic regions. In fact, 96% of the antisense mRNAs were intronic in origin with 64% of these originating from a single 119 kb gene (*CG42339*), suggesting the presence of unannotated ncRNAs in this region. The remaining chromatin-associated RNA detected in our assay arose from non-protein coding transcripts (see *Figure 1—figure supplement 7*), of which 18% were small nucleolar RNA (snoRNA) and 19% were small nuclear RNA (snRNA).

ChAR-seq generated RNA-to-DNA contacts can be aggregated (*Figure 1C*, see e.g., Total RNA), grouped by RNA class (*Figure 1C*, see e.g., mRNA) or viewed individually (*Figure 1C*). Individual RNAs mapped by ChAR-seq generally fell into one of three classes. In the first class, RNAs were found around the locus from which they are transcribed (*Figure 1C*, see, e.g., *Hsromega*, *chinmo*, *ten-m*). In the second class, RNAs were found bound to chromatin in trans, generally distributed across most or all of the genome, often in addition to a peak around the gene body from which the RNA is transcribed (*Figure 1C*, see e.g., *snRNA:U2, snRNA:7SK*). In a third class, RNAs that are part of the dosage compensation complex (*Figure 1C*, *see roX1* and *roX2*) were enriched on and coat the X chromosome. To investigate this first class of RNAs, we compared aggregated RNA-to-DNA contacts with data from nascent transcription sequencing using PRO-seq (*Kwak et al., 2013*), and observed qualitative agreement between PRO-seq and ChAR-seq data sets (*Figure 1D*, *see PRO-seq* and *mRNA*). Nevertheless, many RNA-to-DNA contacts in our dataset are associated in trans to genomic regions outside of the gene body from which the RNA is transcribed (*Figure 1—figure supplement 8*).

To determine if the ChAR-seq protocol disrupts genome organization, we omitted the bridge to produce a mock-treated sample. We next biotinylated the DpnII-digested ends in our mock-sample and ligated them, essentially preparing a Hi-C library ('Hi-C/Mock-ChAR') (*see* Materials and methods for details) from the mock-treated sample (*Figure 1E*, *bottom*). Topologically associated domains (TADs) are preserved in the mock-treated sample and the DNA-DNA contacts show high correlation with previously published CME-W1-cl8+ cell Hi-C (*Ramírez et al., 2015*) (*Figure 1—figure supplements 9–10*). Thus, the three-dimensional genome organization is largely preserved in our protocol. Furthermore, the profile of RNA-to-DNA contacts detected by ChAR-seq was distinctly different from that of DNA-to-DNA contacts in our mock-treated Hi-C library (*Figure 1E* and *Figure 1—figure supplement 9*), indicating that ChAR-seq signal is not simply a byproduct of DNA-DNA contacts.

ChAR-seq data can also be visualized in a two-dimensional contact plot, where the genomic locus from which the RNA is transcribed is represented on the y-axis in linear genome coordinates, and the x-axis defines the genomic location where each RNA was bound. These plots provide a useful overview visualization for of the entire dataset. When we generated these contact plots for ncRNA (*Figure 2A*), mRNA (*Figure 2B*) and snRNA (*Figure 2C*), we observed strong horizontal lines that represent RNA transcripts that are transcribed from a single locus but are found distributed throughout the genome (class II), or in the special case of *roX1* and *roX2*, specifically along the X chromosome (class III). Furthermore, RNAs found at sites from which they are transcribed clustered tightly along the diagonal, a feature most pronounced for mRNAs (class I) (*Figure 2B*). Many of the RNAs we found distributed broadly across the genome are transcription associated small nuclear RNAs (snRNAs) (*Figure 2C*). One of these, *snRNA:7SK*, is an abundant snRNA that functions as a scaffold for a transcriptional regulatory ribonucleoprotein complex that includes p-TEFb, Hexim and LARP7. Other broadly distributed snRNAs are components of the spliceosome (e.g., *snRNA:U2*) which largely functions co-transcriptionally (*Perales and Bentley, 2009*).

To identify RNAs that are highly enriched for chromatin interactions, we plotted the normalized cumulative distribution of the number of sense contacts observed for each gene (*Figure 2D*). The majority of the RNAs in our dataset (14,860 out of 16,812, 88%) had fewer than 10 contacts per kilobase million reads (CPKM) (*Figure 2D*) and were excluded from further analysis. The remaining 1952 RNAs (12%) accounted for 83% (18.5 million) of all chromatin contacts in our data set. To estimate the contribution of total RNA abundance to this interaction signal, we performed RNA-seq for the CME-W1-cl8+ cell line and compared RNA expression levels with RNA-to-DNA contacts identified by ChAR-seq (*Figure 2E*, *Supplementary file 1*). We observed a correlation between RNA expression level and chromatin-RNA contacts; however, a cluster of RNAs clearly generated more chromatin interactions that would be expected from the overall expression levels (*Figure 2E*). Using both the length and read normalized contacts (CPKM) and the fold-enrichment over RNA expression as measured by RNA-seq, we identified 138 RNAs that had more than 100 CPKM and were enriched more than ten-fold, though many were enriched by 2–5 orders of magnitude (*Figure 2E*, red symbols; *Figure 2—figure supplement 1*). Notably, we observe good concordance between the RNAs identified using this methodology between replicates (*Figure 2—figure supplement 2*).

We developed ChAR-seq using the male WME-cl8+ line, reasoning that the ncRNAs *roX1* and *roX2* would serve as an internal positive control. Both *roX1* and *roX2* are part of the MSL2 complex,

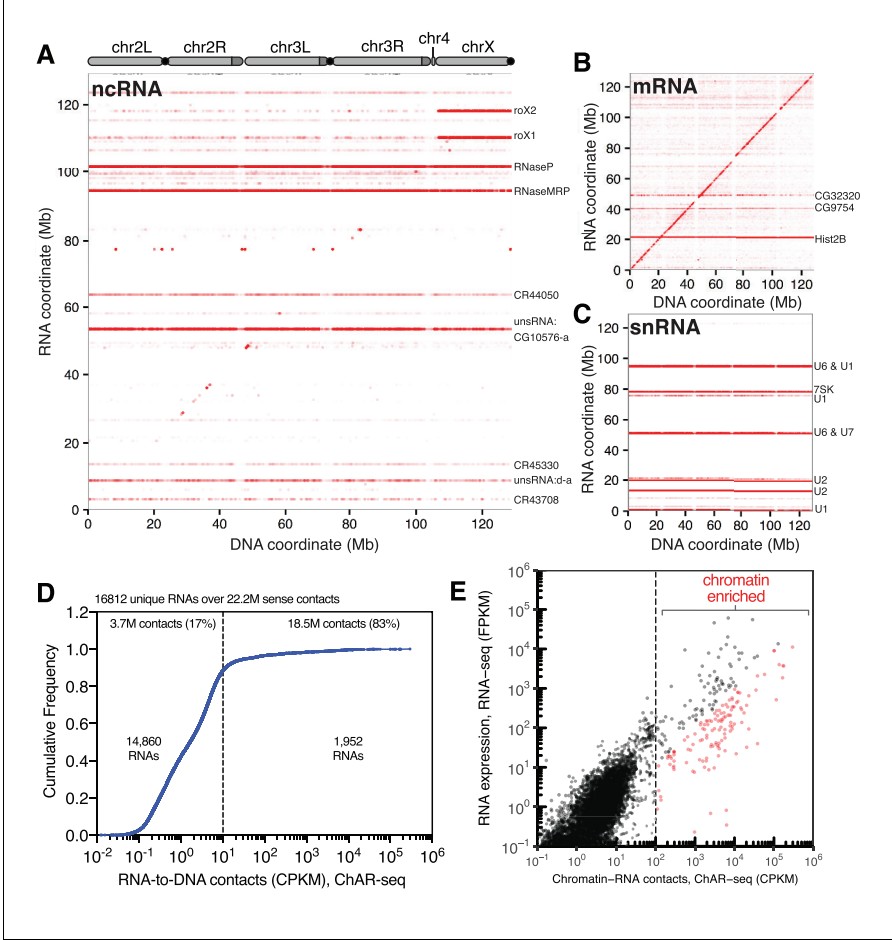

**Figure 2.** ChAR-seq is an 'all to all' RNA-to-DNA proximity ligation method. (**A**) Genome-wide plot of RNA to DNA contacts for non-coding RNAs. The y-axis represents the region of the genome from which a given RNA was transcribed and the x-axis represents the region of the genome where each RNA was found to be associated through proximity ligation (i.e., the binding site for each RNA). Genome-wide contact plots generated in the same way for (**B**) mRNA, and (**C**) snRNA. (**D**) Cumulative frequency of length-normalized contacts for 16,812 RNAs identified on the 'RNA-side' of chimeric reads. The majority (88%) of RNAs have fewer than 10 contacts per kilobase per million reads (CPKM) in our dataset and were not further analyzed owing to low coverage. The remaining 1952 RNAs account for 18.5 million (83%) of the total RNA-to-DNA contacts. (**E**) Scatter plot of length normalized chromatin-contacts versus total expression for each RNA. The 138 RNAs that had more than 100 CPKM and were enriched more than ten-fold are highlighted in red.

DOI: https://doi.org/10.7554/eLife.27024.013

The following figure supplements are available for figure 2:

**Figure supplement 1.** Chromatin-associated RNAs ranked according to CPKM and enrichment in ChAR-seq over RNA-seq.

DOI: https://doi.org/10.7554/eLife.27024.014

**Figure supplement 2.** Chromatin associated RNAs identified by ChAR-seq are highly reproducible.

DOI: https://doi.org/10.7554/eLife.27024.015

which binds across the X-chromosome in male flies to recruit chromatin-modifiers that increase transcriptional output (*Figure 3A*) (*Lucchesi and Kuroda, 2015*). Indeed, ChAR-seq data showed *roX1* and *roX2* to be 7.6-fold (p-value<$10^{-10}$) and 8.1-fold (p-value<$10^{-10}$) enriched for interactions on the X chromosome, respectively (*Figure 3B,C*, *Figure 3—figure supplement 1*). In contrast, female flies express *Sex lethal* (*Sxl*), which binds to *msl2* mRNA to prevent its translation, blocking assembly of the MSL2 complex (*Lucchesi and Kuroda, 2015*). Importantly, *roX1* and *roX2* require MSL2 for X-chromosome specific localization (*Lucchesi and Kuroda, 2015*), therefore female cells should lack

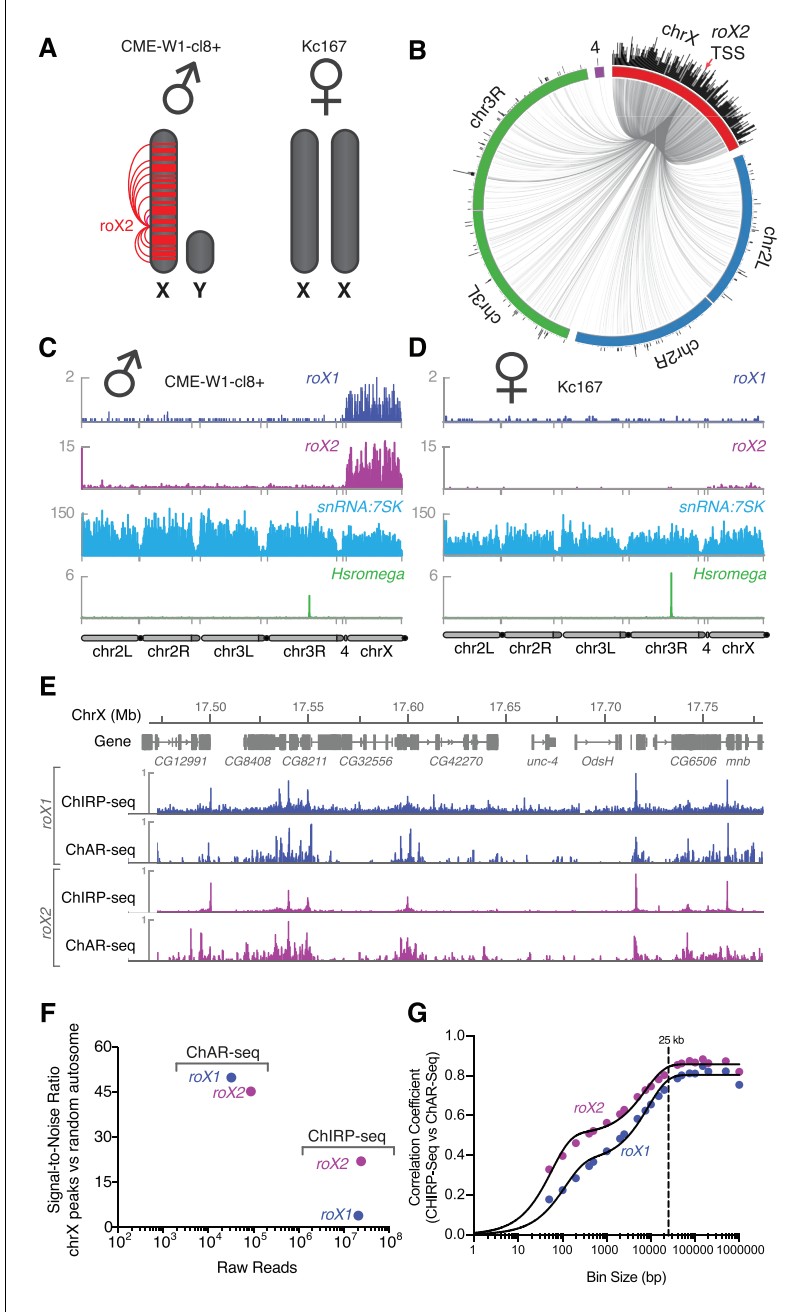

**Figure 3.** Mapping *roX1* and *roX2* of the X chromosome dosage compensation complex. (**A**) Illustration of the *roX1/roX2* spreading across the solitary X chromosome in male flies (CME-W1-cl8+ cell line). In contrast, the female-derived Kc167 cell line expresses significantly lower levels of the MSL2 complex, which mediates the association of *roX1* and *roX2*, which therefore do not coat either of the two X-chromosomes in females. (**B**) Circos plot showing *roX2* spreading from its site of transcription (red arrow) and binding with high density along the X-chromosome but low density binding throughout the genome. (**C**) Coverage plots of *roX1* (blue), *roX2* (purple), *snRNA:7SK* (cyan) and *Hsromega* (green) in male CME-W1-cl8+ cells. Tracks are DpnII normalized reads. ChAR-seq data were subsampled to match the read depth of the Kc167 sample. (**D**) Complementary coverage plots generated from female Kc167 cells. (**E**) Comparison of ChAR-seq (*this work*) to an alternative RNA-to-chromatin mapping method called ChIRP-seq (data from reference [*Quinn et al., 2014*]). Tracks for *roX1* (upper, blue) and *roX2* (lower, purple) were generated from 32308 and 87453 contacts, respectively, from a ChAR-seq dataset containing a total of 22.2 million contacts. For comparison, the *roX1* and *roX2* tracks derived from ChIRP-seq each represent greater that 20 million reads. To compare tracks at different read depths, the contact number was autoscaled, with the maximum peak height given a value of 1. (**F**) Comparison of the signal-to-noise ratio (see

*Figure 3 continued*

methods) between ChAR-seq and Chirp-seq for *roX* genes. 'Raw reads' is the number of *roX* reads present in each data set analyzed. (G) Correlation coefficients were calculated for *roX1* and *roX2* coverage tracks generated using ChIRP-seq and ChAR-seq and plotted relative to increasing bin size to estimate the resolution of the ChAR-seq assay.

DOI: https://doi.org/10.7554/eLife.27024.016

The following figure supplements are available for figure 3:

**Figure supplement 1.** *roX* RNAs are enriched on the X chromosome.

DOI: https://doi.org/10.7554/eLife.27024.017

**Figure supplement 2.** Comparison of RNA-to-DNA contacts between cell type.

DOI: https://doi.org/10.7554/eLife.27024.018

detectable spreading of these ncRNAs along the X-chromosome. When we performed ChAR-seq in a female *Drosophila melanogaster* cell line, Kc167, we did not detect any significant *roX2* localization on the X chromosome (*Figure 3D*) but observed excellent agreement in interaction signal from other RNAs across both cell lines (*Figure 3—figure supplement 2*, *Figure 3C male*, *CME-W1-cl8+* and *Figure 3D*, *female*, *Kc167*, see e.g., *snRNA:7SK* and *Hsromega*).

High-resolution maps of *roX1* and *roX2* localization have previously been generated using ChIRP-Seq, which hybridizes probes against a known RNA and pulls down the associated chromatin for sequencing (*Chu et al., 2011*; *Quinn et al., 2014*). Comparing ChIRP-seq to ChAR-seq for both *roX1* and *roX2* (*Figure 3E*), we found that DNA contact locations were in surprisingly good agreement despite the fact that ChAR-seq reads are spread across all RNAs while ChIRP-seq reads map the specific RNA target, resulting in a large disparity in the effective sequencing depth between the methods. In ChIRP-seq, virtually all of the signal is attributable to interactions between chromatin and the target RNA. In contrast, ChAR-seq captures *all* RNA and DNA contacts, so that any given target RNA will comprise a subset of the total RNA-chromatin contacts in the dataset. In the case of *roX1* and *roX2*, we observed 32,308 and 87,453 contacts, representing 0.1% and 0.36% of the ChAR-seq dataset. In contrast, the ChIRP-seq datasets plotted in *Figure 3E* represent ~24M and ~21M reads for *roX1* and *roX2*, respectively. This indicates that ChAR-seq can identify RNA peaks along chromatin with high sensitivity for a given RNA.

To more quantitatively compare ChAR-seq to ChIRP-seq, we compared the signal-to-noise ratio (SNR) for *roX1* and *roX2* binding to DNA in each assay (*Figure 3F*). We defined signal as the most densely bound regions on chrX, whereas we defined noise as binding events distributed randomly on autosomes (see Materials and methods for details). We found that ChAR-seq *roX1* SNR was much higher than that of ChIRP-seq (49.8 vs 3.9), and ChAR-seq *roX2* SNR was about two-fold higher than that of ChIRP-seq (45.2 vs 22.0), despite having 2 orders of magnitude fewer reads in ChAR-seq. Altogether, these data suggest that ChAR-seq has excellent sensitivity and sufficient signal-to-noise to characterize accurate chromatin-binding events for individual RNAs.

The resolution with which we can measure the localization of an RNA to a given genomic site constrains our ability to assess its potential modes of action. To measure the accuracy of ChAR-seq measurements of RNA interaction with DNA, we compared ChAR-Seq data to ChIRP-seq data to estimate the base-pair resolution of ChAR-Seq. We expected this resolution to be bounded —in part—by the local DpnII cut frequency and the number of contacts for any given RNA. We divided the X chromosome into evenly sized bins and calculated correlation coefficients between ChIRP-seq and ChAR-seq datasets at increasing bin sizes for both *roX1* and *roX2* (*Figure 3G*). Using this method, we noted a bi-phasic increase of the correlation coefficient, corresponding to a minor plateau around 200 bp and a major plateau at ~25 kbp. The minor plateau is likely due to the DpnII distribution bias in the ChAR-seq tracks, while the major plateau is an estimate of the resolution of our assay, which is on the order of other proximity-ligation sequencing assays like Hi-C (*van Berkum et al., 2010*).

To test if we could identify the functional roles for our most highly enriched RNAs, we clustered the snRNA class of RNAs based on their genomic contacts. These snRNAs collectively comprised 23% of all the RNA-to-DNA contacts in our dataset (*Figure 4A*) and are a substantial component of the spliceosome, a multi-megadalton ribonucleoprotein complex that catalyzes pre-mRNA splicing (*Zhou et al., 2002*; *Will and Lührmann, 2011*). The composition and conformation of the

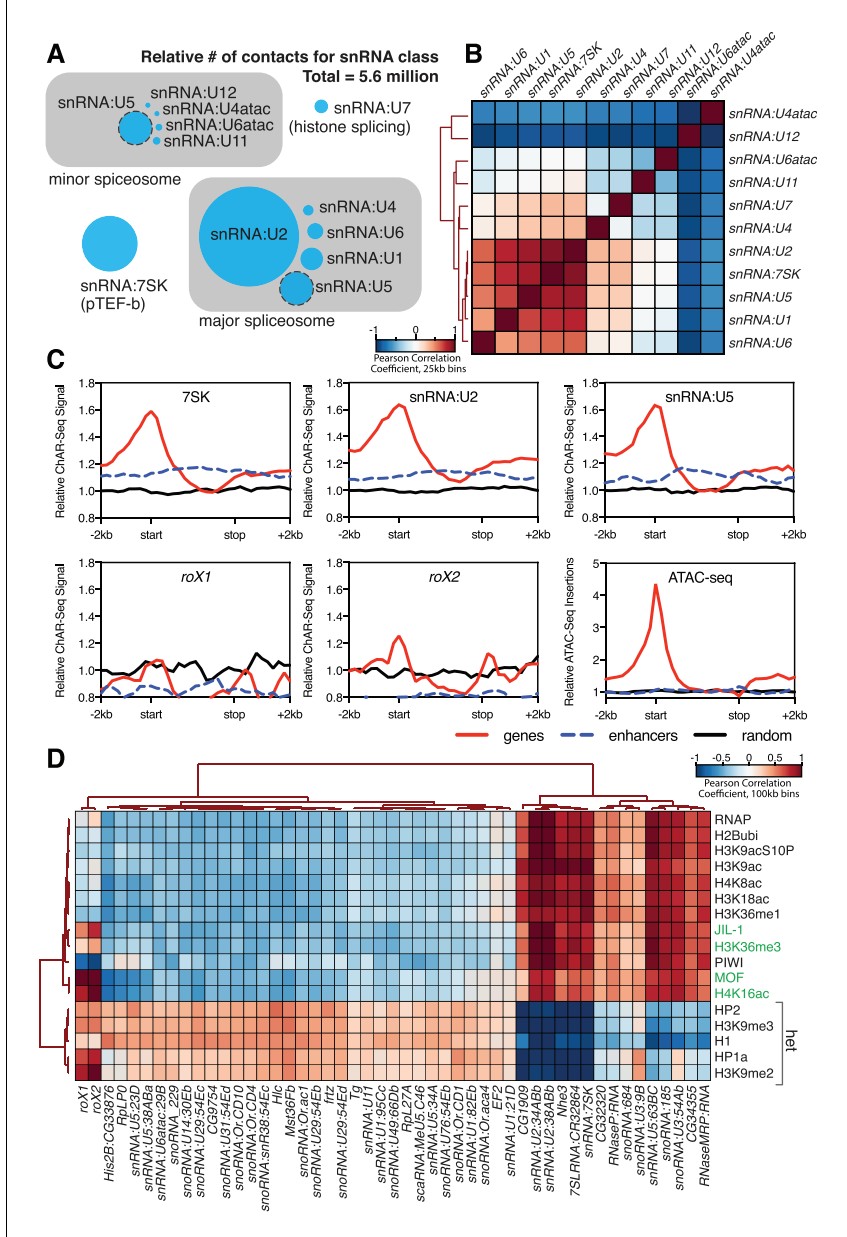

**Figure 4.** Correlation of chromatin-associated RNAs with genome features. (**A**) Relative abundance of snRNAs identified by ChAR-seq. The size of the circles is proportional to the abundance of the snRNAs found by ChAR-seq. RNA components of the major and minor spliceosome are bounded by the gray boxes. (**B**) Cluster analysis of the pairwise correlation between genome-wide tracks of snRNAs. (**C**) Meta-analysis plots aggregating the signal of *snRNA:7SK*, *snRNA:U2*, *snRNA:U5*, *roX1*, *roX2* and ATAC-seq over gene bodies (red), putative enhancers (blue dashed line) and random regions (black). (**D**) Hierarchical clustering based on pairwise Pearson correlation between representative ChAR-seq RNA-to-DNA contact coverage tracks (black) and modENCODE datasets available for the WME-C1-cl8 + cell line. Notable associations for the dosage compensation complex (green) and heterochromatin ('het') are indicated in the right margin.
DOI: https://doi.org/10.7554/eLife.27024.019

The following figure supplements are available for figure 4:

**Figure supplement 1.** Sequence similarity of snRNAs.
DOI: https://doi.org/10.7554/eLife.27024.020

**Figure supplement 2.** RNA abundance over TAD boundaries and centers.
DOI: https://doi.org/10.7554/eLife.27024.021

spliceosome is highly dynamic, though two dominant species exist in eukaryotes: the major spliceosome comprised of *U1*, *U2*, *U4*, *U5* and *U6* snRNAs, and the minor spliceosome comprised of *U4:atac*, *U6:atac*, *U5*, *U11*, and *U12* (*Will and Lührmann, 2011*). Many members of this class of snRNAs have highly similar gene duplication variants in the *Drosophila* genome. We therefore first calculated the base sequence similarity of these variants to one another and aggregated signals that were tightly clustered (*Figure 4—figure supplement 1*). When we then correlated genome-wide binding signal within this class, we found that the distribution patterns of the major spliceosome snRNAs *U1*, *U2*, *U4*, *U5*, *U6* clustered together along with *snRNA:7SK* (*Figure 4B*), which is part of the p-TEFb complex that relieves pausing of RNA Polymerase II at promoters (*Kwak and Lis, 2013*) and may participate in the release of paused polymerase during RNA splicing (*Barboric et al., 2009*). The components of the minor spliceosome did not cluster together, likely due to their low abundance (*Will and Lührmann, 2011*) and consequently low representation in our dataset.

We next reasoned that spliceosome RNAs—as part of the co-transcriptional RNA processing machinery—should also be enriched in regions of active transcription. We therefore aggregated spliceosomal RNA signals over gene bodies (*Figure 4C*, *red lines*), putative enhancers (*Kvon et al., 2014*) (*Figure 4C*, *blue dashed lines*) and a random distribution of genomic bins of similar size (*Figure 4C*, *black lines*). We observed an enrichment of snRNAs (*7SK*, *U2* and *U6*), but not *roX1* or *roX2*, over gene bodies (*Figure 4C*) with a broad peak around transcription start sites, in good agreement with ChIRP data for *7SK* in mice (*Flynn et al., 2016*). Because active transcription is also correlated with topological boundaries in flies (*Hou et al., 2012*; *Ulianov et al., 2016*), we examined the relationship between RNA contacts and genome organization. To test whether RNA-DNA contacts are enriched at topological boundaries, we measured the DNA contact frequency for *snRNA:7SK* and *snRNA:U2* in 20 kb windows spanning TAD boundaries and TAD centers. Both *7SK* and *U2* RNAs were modestly, but significantly (Wilcoxon rank-sum test) enriched at TAD boundaries (*Figure 4—figure supplement 2A* (top)). We also aggregated *7SK* and *U2* contact signals across all TAD boundaries (*Figure 4—figure supplement 2A* (bottom), *red lines*) and a random distribution of identically sized genomic bins (*Figure 4—figure supplement 2A* (bottom), *black lines*), and found that these RNAs were ~1.4 fold enriched over boundaries. We also noted that *roX2* was ~2.2 fold enriched over TAD boundaries on the X chromosome (*Figure 4—figure supplement 2B*), consistent with previous data showing that MSL2 dosage compensation complex High Affinity Sites (HAS) are preferentially found at TAD boundaries (*Ramírez et al., 2015*). Examination of chromatin accessibility (ATAC-seq) over TAD boundaries (*Figure 4—figure supplement 2B*) showed that open chromatin is enriched at TAD boundaries, supporting the idea that TAD boundaries are transcriptionally active in flies.

In contrast to the small number of well-defined and well-characterized snRNAs involved in splicing, there are more than 200 snoRNAs in flies (*Huang et al., 2005*) that are significantly divergent in sequence and, surprisingly, were highly represented in our dataset (*Figure 1—figure supplement 7* and *Figure 2—figure supplements 1–2*). Most of these snoRNAs have either unknown function or are computationally identified and indirectly implicated in the maturation and modification of ribosomal rRNA (*Huang et al., 2005*).

To determine if our enriched chromatin-associated RNAs, in particular snRNAs and snoRNAs, might localize to euchromatic or heterochromatic states or with specific transcription factors, we cross-correlated our ChAR-seq signal against modENCODE datasets available for the CME-W1-cl8 + cell line. To normalize the signals for comparison, we first calculated the expected contacts per 2 kb bin for each RNA under a uniform distribution, based on the total number of genome-wide contacts for each RNA and the number of DpnII sites per bin. This null model was then used to calculate the log2 ratio of the observed to the expected contacts per bin for each RNA, which was then transformed into a z-score ($(x-\mu)/\sigma$) based on the whole-genome mean ($\mu$) and standard deviation ($\sigma$). Similarly, we re-binned the modENCODE tracks, removed bins that did not contain a DpnII site, and transformed the log2 mean-shift values to a z-score. We then calculated the pair-wise Pearson correlation coefficients between each signal track, and then clustered the data (*Figure 4D*). We observed discrete clustering of *roX1* and *roX2* with known dosage compensation complex factors, MOF, the histone modifications H4K16ac and H3K36me3 (*Bell et al., 2008*), and JIL-1 kinase (*Bai et al., 2004*), validating this analytical approach (*Figure 4D*). Beyond this sub-cluster of dosage compensation factors, the remainder of the chromatin-associated RNAs fell into two distinct and anti-correlated categories: those associated with active chromatin and transcription (e.g., RNAPII, H4K8ac, H3K18ac) or

heterochromatin (e.g., HP2, H3K9me3, HP1a) (*Figure 4D*). In particular, we note that *snRNA:U2* and *snRNA:7SK* cluster tightly with the transcription-associated chromatin marks, while many of the snoR-NAs and minor spliceosome snRNAs that we identified are associated with heterochromatin, likely due to co-localization of heterochromatin factors to the nucleolus. Interestingly, *snRNA:U5*, a component of both the major and minor spliceosome, has variants that clearly cluster with either transcriptionally active chromatin (63BC) or heterochromatin (23D, 38ABa, and 34A). Previous work has shown that the *snRNA:U5:38Aba* variant (*Figure 4D*, heterochromatin cluster) exhibits a unique tissue-specific expression profile with the greatest abundance in neural tissue, which led the authors to propose isoform-dependent functions in alternative splicing (*Chen et al., 2005*). The differential clustering that we observe for *snRNA:U5*, and between major and minor spliceosome snRNAs, for euchromatin and heterochromatin might reflect such isoform-specific functions of the spliceosome in different chromatin states.

## Discussion

ChAR-seq maps the chromosomal binding sites of all chromatin-associated RNAs, independent of whether they are associated as nascent transcripts or bound as part of ribonucleoprotein complexes (RNPs). In this way, ChAR-seq can be thought of as a massively parallelized de novo RNA mapping assay capable of generating hundreds to thousands of RNA-binding maps. ChAR-seq also detects multiple classes of chromatin-associated RNAs. ChAR-seq preserves the three-dimensional structure of the genome and provides an RNA-DNA interaction matrix that complements DNA-DNA proximity measurements using Hi-C. We validated ChAR-seq using chromosome-specific ncRNAs *roX1* and *roX2* associated with dosage compensation. The comparison between ChAR-seq and ChIRP-seq, which vary dramatically in the sequencing depth needed to analyze a specific RNA, highlights the utility of ChAR-seq as a de novo chromatin-associated RNA discovery tool. ChAR-seq also maps nascent RNAs found at the loci from which they are transcribed.

ChAR-seq is similar to a recently published method (*Sridhar et al., 2017*), but has two key distinctions. First, proximity ligations are performed in situ in intact nuclei, which reduces nonspecific interactions (*Rao et al., 2014*). Second, ChAR-seq uses long single-end reads to sequence across the entire junction of the 'bridge', ensuring that RNA-to-DNA contacts are mapped with high confidence and reporting on the polarity of the bridge-ligated RNA.

While this work was in revision, a genome-wide RNA-DNA contact method—GRID-seq—was published that also uses proximity ligation of a directional bridge to RNA and DNA (*Li et al., 2017*). One key difference is that GRID-seq uses a restriction enzyme to cut cDNA and gDNA fragments 19–23 bp distal to the bridge, which allows size selection of molecules that contain both RNA and DNA. This likely reduces the number of uninformative molecules sequenced. A disadvantage, however, is that the resulting reads are 19–23 bp, which have a greater chance of falsely mapping during genome alignment than the 20–100 bp RNA and DNA fragments obtained with ChAR-seq. Although their details differ, ChAR-seq and GRID-seq appear to have similar ability to detect *roX2* binding to chrX in *Drosophila* (see *Figure 3—figure supplement 1B*).

We used ChAR-seq to discover and map several dozen ncRNAs that are pervasively bound across the genome. Many of these ncRNAs are components of ribonucleoprotein complexes associated with transcription elongation (*snRNA:7SK*), splicing (*snRNA:U2*, etc) and RNA processing (*snRNAs, snoRNAs* and *scaRNAs*). Interestingly, more than half of the chromatin-associated RNAs identified based on our enrichment criteria are snoRNAs, most of which—but not all—correlate with heterochromatin. Generally, snoRNA ribonucleoproteins (snoRNPs) use intermolecular base pairing to direct chemical modification of the 2'-hydroxyl groups or the isomerization of uridines to pseudouridine (*Cech and Steitz, 2014*) and snoRNAs are known abundant components of chromatin in both flies (*Schubert et al., 2012*) and in mice (*Meng et al., 2016*). Despite their abundance and the their known role in RNA modification, we do not yet understand the functions of these modifications, or the implication of snoRNAs and scaRNAs chromatin association in cells (*Cech and Steitz, 2014*). We also demonstrate that ChAR-seq can be used with orthogonal genome-wide datasets to identify and classify RNAs that are associated with specific chromatin states (e.g., euchromatin vs heterochromatin). We expect this approach will be particularly useful in organisms that use lncRNAs such as *HOTAIR*, *HOTTIP* and *BRAVEHEART* as scaffolds for ribonucleoproteins that regulate facultative heterochromatin. Finally, we observed enrichment of transcription-associated RNAs with TAD

boundaries, reflecting a potential role for active transcription in the topological organization of the genome. This observation is consistent with previous observations that active transcription is a stronger predictor for TAD partitioning in flies (*Hou et al., 2012*; *Ulianov et al., 2016*) than CTCF and cohesin, the prototypical TAD boundary markers in mice and humans (*Merkenschlager and Nora, 2016*).

We anticipate that ChAR-seq will be a powerful new high throughput discovery platform capable of simultaneously identifying new chromatin-associated RNAs and mapping their chromatin binding sites (and associated epigenetic chromatin states), all of which will be particularly useful in comparing 'epigenomic' changes that coincide with cellular differentiation or tumorigenesis.

## Materials and methods

### Brief description of ChAR-Seq protocol

*Drosophila melanogaster* CME-W1-cl8+ cells (Drosophila Genome Resource Center, Stock #151) were grown in T-75 flasks at 27°C in Shields and Sang M3 media supplemented with 5 µg/mL insulin, 2% FBS, 2% fly extract and 100 µg/mL Pen-Strep (*Cherbas et al., 2011*). Approximately 100–400 million cells were harvested for each library by centrifugation at 2000 x g for 2–4 min, resuspended in fresh media plus 1% formaldehyde and fixed for 10 min at room temperature. Fixation was quenched by adding 0.2 M glycine and mixing for 5 min at room temperature. Cells were centrifuged at 2000 x g for 2–4 min, resuspended in 1 mL of PBS, and centrifuged again at 2000 x g for 2 min. The supernatant was aspirated and discarded, and the cell pellet was flash frozen in liquid nitrogen and stored at −80C until needed. Cells were thawed in lysis buffer and the cross-linked nuclei and cellular material were isolated by centrifugation for the in situ ligation protocol. Female Kc167 cells (Drosophila Genome Resource Center, Stock #1) for analysis of sex dependent RNA contact map were processed as for CME-W1-cl8+ cells. Both Kc167 and CME-W1-cl8+ cells were authenticated by the Drosophila Genome Resource Center and tested for mycoplasma contamination by cytoplasmic DNA staining.

Briefly, RNA was lightly and partially chemically fragmented by heating in the presence of magnesium. The pellet was isolated and washed, and RNA ends were ligated using truncated T4 Rnl2tr R55K K227Q ligase (hereafter referred to as trT4KQ RNA ligase) to an oligonucleotide 'bridge' molecule containing a 5′-adenylated ssDNA overhang. The RNA ligase was inactivated, the pellet was washed and the RNA strand was stabilized by first strand synthesis of the RNA through extension of the bridge by Bst 3.0 polymerase. The polymerase was inactivated and the pellet was washed. Genomic DNA was then digested with DpnII, followed by ligation of the DpnII digested genomic DNA to the opposite end of the oligonucleotide 'bridge'. Second strand synthesis was then performed using RNaseH and DNA Polymerase I to complete cDNA synthesis of the RNA-encoded side of the new, chimeric molecules. The sample was then deproteinized and crosslinks were reversed by heating overnight in SDS and proteinase K. DNA was then ethanol precipitated and sheared to ~200 bp fragments using a Covaris focused ultra-sonicator. DNA fragments containing the biotinylated bridge were then purified using magnetic streptavidin-coated beads. DNA ends were repaired using the NEBNext End Repair and dA tailing module, and ligated to NEBNext hairpin adaptors for Illumina sequencing. The adaptor hairpin was cleaved using USER, and DNA fragments were amplified by ~8–12 rounds of PCR with NEBNext Indexing Primers for Illumina (TruSeq compatible). The partially amplified library was then purified using AMPure XP beads to remove adaptor dimers, and the optimum number of additional PCR cycles was determined by qPCR to achieve approximately 30% saturation. Library amplification was then completed by the additional rounds of PCR, and the library was purified and size selected to a target range of 100–500 bps using AMPure XP beads. The size distribution of the library was checked by capillary electrophoresis using an Agilent Bioanalyzer, and quantified using qPCR against a phiX Illumina library standard curve. Libraries were sequenced using the Illumina MiSeq platform for quality control, and subsequently sequenced on the Illumina NextSeq platform (Stanford Functional Genomics Facility) using single-end 152 bp reads. Data were processed and analyzed using a custom pipeline.

## Detailed ChAR-seq protocol
## Shields and Sang M3 media, for 0.5 L

- 19.6 g S and S M3 powder (Sigma)
- 0.25 g bicarb
- 5 µg/mL insulin
- 2% FBS
- 2% fly extract
- 1x Pen/Strep

Sterile filter media using 0.2 um filter. Store media at 4C. Warm to room temperature before splitting cells. Culture cells at 27C in 75 flasks (1–2 flasks per library), splitting or harvesting approximately every 3–4 days. Cells are detached from the flask surface by vigorously pipetting up and down with a serological pipette. Confirm that >95% of cells are healthy and viable using trypan blue assay at each harvest.

All buffers and solutions prepared with DEPC-treated water. All buffers are made with fresh, unopened chemicals to minimize RNase contamination. All enzyme stocks and reagents are dedicated for RNA work and kept RNase-free. Optional steps are included for the three alternative protocols used for (A) the RNA spike-in experiment, (B) the RNase control, and (C) the Hi-C/Mock-ChAR experiment.

## Step 1: Harvesting cells and crosslinking (can be done in advance)

- Pipette cells up and down to detach them from the surface. The cells grow in small strings or clumps that are very weakly attached to the surface.
- Count the cell using a hemocytometer. Typically, each T75 flask should yield between 100–150 M cells.
- Spin (~2000 x g) the cells in either one or two 50 µL conical tubes in a swinging bucket centrifuge for 5 min.
- Resuspend the cells in ~36 mL of fresh media
- Add formaldehyde to 1% (1 mL of 37% form)
    - incubate at RT for 10 min mixing end over end
- Quench by adding 3 mL of 2.5 M glycine (final concentration = 0.2 M), mix for 5 min end over end
- Spin the cells at 2000 x g for 5 min
- Resuspend in ~1 mL of sterile filtered, ice cold PBS + DEPC
    - count cells and split into an appropriate number of tubes to aliquot 100 M cells per tube
- Spin again at 2000 x g for 5 min
- Remove supernatant and flash freeze in liquid nitrogen

NOTE: we find it useful to remove the supernatant in this and all subsequent steps using a mechanical micropipette rather than an aspirator, which risks loss of sample.

## Step 2: Lysis (Day 1, start in early afternoon)
## Lysis buffer

- 10 mM TrisHCl pH 8
- 10 mM NaCl
- 0.2% igepal
- 1 mM DTT
- 0.2 mM EDTA
- Mix 750 µL lysis buffer +150 µL Sigma protease inhibitor (P8340, comes as a DMSO stock)+45 µL RNaseOut per sample.
- Add 400 µL lysis buffer plus inhibitors to STILL FROZEN crosslinked cells and thaw with gentle mixing. Incubate for 2 min on ice after thawing is complete. _DO NOT let cells thaw on ice before adding lysis buffer!_
- Centrifuge at 2.5 k x g for 2 min, discard supernatant.
- Wash pellet with 500 µL of lysis buffer +inhibitors, spin again for 1 min.
- Gently resuspend the pellet in 100 µL 0.5% SDS and incubate at 62C for 5 min
- Pre-mix 290 µL water +50 µL 10% Triton-X 100 per sample.

- Quench SDS with diluted Triton-X 100. Mix well by gentle pipetting; incubate at 37C for 15 min.

## Step 3: RNA fragmentation (Day 1)

- Add 11 μL 10x T4 RNA ligase buffer for final concentration 0.25x
- Incubate for <u>exactly 4 min</u>, 70'C
- Snap cool on ice
- Note: Users may want to optimize fragmentation beforehand if using a different cell line or sample.

## Step 3A (Optional additional step A): RNA spike in

- Estimated RNA release is 65 μg / 250 M cells
- For 180 M cells, spike in 5.6, 0.56 and 0.056 μg
  - 1.86 μL each of 100%, 10% and 1% RNA stocks of MBP RNA
  - 1.55 μL each of 100%, 10% and 1% RNA stocks of GFP RNA
  - 1.55 μL each of 100%, 10% and 1% RNA stocks of Halo RNA
- These are designated as 'high', 'med' and 'low'
- Incubate on ice for 5 min

## Step 4: Wash cells to remove SDS and non-crosslinked RNA fragments (Day 1)

- Add 1000 μL DEPC-treated PBS and incubate on ice for 2 min
- Centrifuge at 2.5 k x g for 2 min, discard supernatant
- Add 1000 μL DEPC-treated PBS, mix gently
- Centrifuge at 2.5 k x g for 2 min, discard supernatant
- Immediately proceed to the next step, with pre-mixed reaction buffer already prepared

## Step 4B (Optional additional step B): RNase control

- Pre-mix and resuspend the pellet in the following
  - 160 μL water
  - 20 μL 10x T4 RNA ligase buffer
  - 10 μL 10 mg/mL RNaseA
  - 10 μL RNaseH (NEB)
- Incubate at 37C for 4 hr
- Centrifuge at 2.5 k x g for 2 min, discard supernatant
- Add 1000 μL DEPC-treated PBS, mix gently
- Centrifuge at 2.5 k x g for 2 min, discard supernatant
- Immediately proceed to the next step, with pre-mixed reaction buffer already prepared

## Step 5: RNA to linker ligation, RNA first, Day 1

- Pre-mix the following and then add to cells
  - 20 μL 10X T4 trKQ RNA ligase buffer
  - 100 μL 50% PEG
  - 20 μL RNaseOUT
  - 20 μL 50 uM annealed bridge (2.5 nmols)
  - 10 μL T4 trKQ RNA ligase (4000 units)
- Incubate at 23C overnight, shaking 900 rpm in ThermoMixer

## Step 5 (alternative): Hi-C control (PEG mock incubation, no ligase, no bridge)

- Pre-mix the following and then add to cells
  - 20 μL 10X T4 trKQ RNA ligase buffer
  - 100 μL 50% PEG
  - 20 μL RNaseOUT
  - 30 μL water
- Incubate at 23C overnight, shaking 900 rpm in ThermoMixer

End day 1

_________________________________________________________________

Start day 2

### Step 6: Wash to remove PEG (Day 2, morning)

- Centrifuge at 2.5 k x g for 4 min at RT, remove and discard supernatant
- Resuspend pellet with 1 mL DEPC-treated PBS

- Centrifuge at 2.5 k x g for 5 min at room temperature
- While spinning, make 200 µL 1x T4 trKQ RNA ligase buffer, including:
  - 0.2 µL 1 M DTT
  - 10 µL RNaseOUT
- Resuspend pellet in 200 µL 1x T4 trKQ RNA ligase buffer +DTT and RNaseOUT

### Step 7: First strand synthesis, Bst 3.0 (Day 2, morning)

- Add 20 µL 10 mM (each) dNTP mix
- Add 15 µL Bst 3.0 (120 U/ µL) for 1800 Units
- Incubate for 10 min at RT
- Increase temp to 37°C for 10 min (shaking 900 rpm in ThermoMixer)
- Increase temp again to 50°C for 20 min
- Add 1000 µL DEPC-treated PBS, mix, and cool on ice 1 min

### Step 7 (alternative): Hi-C control (mock, no Bst, no NTPs)

- Add 35 µL PBS (no enzyme, no dNTPs)
- Incubate for 10 min at RT
- Increase temp to 37°C for 10 min (shaking 900 rpm in ThermoMixer)
- Increase temp again to 50°C for 20 min
- Add 1000 µL DEPC-treated PBS, mix, and cool on ice 1 min

### Step 8: Inactivate and wash (Day 2)

- Centrifuge at 2.5 k x g for 2 min, discard supernatant
- Repeat wash with another 1 mL of DEPC-treated PBS
- Centrifuge at 2.5 k x g for 2 min, discard supernatant
- Resuspend pellet in 100 µL 0.5% SDS +0.1 mM EDTA
- Heat inactivate for 10 min at 55C
- Quench SDS with 290 µL water + 50 µL 10% Triton-X 100 (pre-mixed), per sample.
- Mix well, incubate at 37°C for 15 min.
- Add 1000 µL DEPC-treated PBS and incubate on ice for 2 min
- Centrifuge at 2.5 k x g for 2 min, remove and discard supernatant.
- Wash nuclear pellet by adding 1 mL DEPC-treated PBS, mix, and spin at 2.5 k x g for 2 min
- Remove and discard supernatant, and proceed IMMEDIATELY to digestion.

### Step 9: Genomic DNA digestions (Day 2, afternoon)

- Resuspend in 200 µL digest buffer: 20 µL 10X T4 RNA ligase buffer + 169.8 µL DEPC-treated water + 0.2 µL 1 M DTT (1 mM DTT final) + 10 µL RNaseOUT (pre-mixed).
- Add 15 µL DpnII (750 Units)
- Incubate at 37°C for at least 3 hr, up to overnight if desired.
- Terminate digestion by adding 7.5 µL 0.5 M EDTA to a final concentration of ~15 mM

### Step 10: Pellet and wash cells

- Spin the cells at 2500 x g for 2 min
- Remove and discard the supernatant
- Re-suspend the X-linked cells with 1 mL DEPC-treated PBS,
- Spin again at 2500 x g for 2 min
- Remove and discard the supernatant using a P200

### Step 11: Inactivate DpnII

- Re-suspend the washed cells in 100 µL 0.5% SDS +0.1 mM EDTA

- Incubate at 55°C for 10 min
- Quench SDS by adding 290 µL water + 50 µL 10% Triton-X 100
- Incubate at 37°C for 5 min
- Add 1 mL of PBS and incubate on ice for 5–10 min

## Step 12: Pellet and wash cells three times

- Spin the cells at 2500 x g for 2 min, discard supernatant
- Re-suspend the pellet with 1 mL DEPC-treated PBS
- Spin again at 2500 x g for 2 min, discard supernatant
- Re-suspend the pellet with 1 mL DEPC-treated PBS
- Spin again at 2500 x g for 2 min, discard supernatant

## Step 12 (alternative): Hi-C control (biotinylated the cut ends)

- Perform washes as in Step 12 above
- Pre-mix the following:
  - 64 µL DEPC water
  - 10 µL 10X T4 RNA ligase buffer
  - 10 µL 0.4 mM biotin-dATP
  - 4 µL 1 mM dCTP
  - 4 µL 1 mM dGTP
  - 4 µL 1 mM dTTP
  - 4 µL Klenow (exo minus)
- Add pre-mixed biotinylation solution to samples
- Incubate at 23°C for 3–4 hr
- Add 4 µL 500 mM EDTA
- Heat inactivate at 60°C for 10 min
- Spin the cells at 2500 x g for 2 min
- Remove and discard the supernatant
- Re-suspend the pellet with 1 mL DEPC-treated PBS
- Spin again at 2500 x g for 2 min

## Step 13: Bridge to genomic DNA ligation (Day 2 overnight to day 3)

- Resuspend pellet in the following pre-mixed solution:
  - 140 µL DEPC-treated water
  - 20 µL RNaseOUT
  - 20 µL T4 DNA ligase buffer
  - 20 µL T4 DNA ligase (8000 Units)

NOTE : this **reaction requires ATP** and this is **different** from the buffer used throughout the earlier steps, if you use T4 RNA ligase buffer, you must supplement the final concentration with 1 mM ATP

- Incubate at 16C overnight

End day 2

_______________________________________________________________________________

Start day 3

## Step 14: Pellet and wash cells, to remove T4 ligase
**\*\* Allow Hi-C ligation to proceed during second strand synthesis, so skip this step for Hi-C sample \*\***

- Add 7.5 µL 0.5 M EDTA (~15 mM final)
- Spin the cells at 2500 x g for 2 min
- Remove and discard the supernatant
- Re-suspend the pellet with 1 mL DEPC-treated PBS
- Spin the cells at 2500 x g for 2 min
- Remove and discard the supernatant
- Re-suspend each pellet in the following pre-mixed solution:
  - 80 µL DEPC-treated water
  - 10 µL 10x T4 RNA ligase buffer

- ○ 1 µL 1 M DTT
- ○ 10 µL RNaseOUT

## Step 15: Second strand synthesis (skip for Hi-C control)

- 2x cDNA bufffer
  - ○ 10 mM TrisHCl (8.0)
  - ○ 180 mM KCl
  - ○ 100 mM $(NH_4)_2SO_4$
- Add 100 µL 2x cDNA buffer
- Add 20 µL 10 mM dNTP (each) mix
- Add 100 Units *E. coli* DNA Polymerase I (NEB)
- Add 5 Units RNaseH (NEB)
- Mix and incubate at 16C for at least two hours

## Step 16: Crosslink reversal (end of Day 3)

- Centrifuge at 2.5 k x g for 5 min
- Remove and discard supernatant
- Resuspend pellet in 400 µL DEPC-treated PBS
- To ~450 µL of sample, add
  - ○ 55 µL 10% SDS
  - ○ 55 µL 5 M NaCl
  - ○ 3 µL 20 mg/mL Proteinase K
  - ○ incubate overnight at 70°C

End day 3
________________________________________________________________

Start day 4

## Step 17: Precipitate DNA

- Cool sample to room temperature

NOTE: Do not cool the sample on ice as the SDS will precipitate

- Add 50 µL 3 M NaOAc
- Add 2 µL (5 mg/ml) glycogen
- Add 850 µL ice cold 100% EtOH.

NOTE: Flocculent should be visible after inverting the tube gently 3–4 times. If no flocculent is visible, add 200 µL of additional ice cold 100% EtOH

- Incubate on ice for 30 min
- Centrifuge at ~21,000 x g for 5–10 min
- Wash pellet with 70% ice cold EtOH
- Centrifuge at ~21,000 x g for 5 min
- Gently dry pellet and resuspend in 130 µL TE

## Step 19: Covaris shear DNA

- Shear DNA to ~200 bp fragments using the following settings: 10% duty factor, 175 peak incident power, 200 cycles per burst, for 180 s
- NOTE: shearing of Hi-C sample should target 300–500 bp fragments... make appropriate adjustments for AMPure steps

## Step 19: Isolation of biotinylated DNA fragments

1X tween wash buffer (TWB)

5 mM TrisHCl (7.5)
0.5 mM EDTA
1 M NaCl
0.05% Tween 20

2X bead binding buffer (BBB)

10 mM TrisHCl (7.5)
1 mM EDTA
2 M NaCl

- Wash 150 μL Dynabeads MyOne SA T1 beads (Life Tech) with 400 μL TWB. Pull down beads with a magnetic tube rack and discard supernatant.
- Resuspend the beads in 130 μL 2X BBB
- Mix the bead slurry with the DNA (final volume of 260 μL)
- Incubate at RT for 15 min with rotation to bind biotinylated DNA
- Pull down beads with a magnetic tube rack and discard supernatant.

NOTE: Do not allow the beads to dry out

- Wash the beads by adding 750 μL 1x TWB and mixing
- Warm the tubes to 50°C for 2 min
- Pull down beads with a magnetic tube rack and discard supernatant
- Wash the beads by adding 750 μL 1x TWB and mixing
- Pull down beads with a magnetic tube rack and discard supernatant

## Step 20: End repair, and dA tailing

- Resuspend the beads in 40 μL TE
- add 7 μL NEB Next End Prep Buffer
- add 3 μL NEB Next End Prep Enzyme Mix
- Incubate at RT for 20 min with agitation to keep beads suspended
- Increase temp to 65°C for 30 min
- Cool to room temperature

## Step 21: Adapter ligation (on bead)
NOTE: The following order of addition is important

- add 2.5 μL NEBNext Adaptor **FIRST**
- add 1 μL ligation enhancer
- add 30 μL NEBNext UltraII Ligation Master Mix **LAST**
- mix vigorously, incubate at **Room Temperature** for 15–20 min with agitation to keep beads suspended
- add 3 μL USER enzyme
- incubate at 37°C for 15 min

## Step 22: Wash beads

- Pull down beads with a magnetic tube rack and discard supernatant
- Wash the beads by adding 750 μL 1x TWB and mixing
- Warm the tubes to 50°C for 2 min
- Pull down beads with a magnetic tube rack and discard supernatant
- Wash the beads by adding 750 μL 1x TWB and mixing
- Pull down beads with a magnetic tube rack and discard supernatant

## Step 23: On bead amplification (library PCR 1)

- For each index, pre-mix the following
  - 30 μL 2x NEB Next High Fidelity master mix
  - 3 μL 10 uM Universal Primer (NEBNext Multiplex Oligos for Illumina)
  - 3 μL 10 uM Indexing Primer (NEBNext Multiplex Oligos for Illumina)
  - 24 μL water
- Resuspend the beads in 50 μL amplification mix (reserve the extra 10 μL as you will need it in step 25)
- Run the following PCR cycle
  - 98°C for 30 s

- ○ 63°C for 30 s
- ○ 72°C for 40 s
- ○ 98°C for 3 min
- ○ then 8 cycles of
- Stop the PCR, pull down beads in a magnetic rack

## Step 24: First AMPure XP size selection to remove adaptor dimers (target size 200 to 500 bp)

NOTE: The NEBNext adaptor is a 65 nt hairpin… adaptor dimers would be 65 bp, because each hairpin is ~32 bp long… however, the indexing primers are also 65 nts, so amplification of adaptor dimer results in a 130 bp fragment that must be removed at this step, and adds 130 bp to the ideal target length of 150 bp, so our molecules of interest are now 280–300 bp

- Allow AMPure bead slurry to warm to room temperature
- Add an equal volume of AMPure beads to the PCR supernatant, mix by pipetting and incubate for 5 min at room temperature
- Pull down beads with magnet, give 1–2 min to clear solution. Remove supernatant.

NOTE: This is, in theory, to be discarded, but save until you've validated the completed library prep by qPCR and Bioanalyzer in case needed for troubleshooting

- Wash the beads with 70% EtOH (e.g., 700 µL), pull down, remove sup, and
- elute with 30 µL of 10 mM Tris pH 8

## Step 25: qPCR to determine how many more cycles to amplify

- Mix 5 µL of each library eluted from the AMPure beads with 10 µL of the reserved PCR mix from step 23
- Dilute 100x Syber Green 1:3 in water for a 33x working solution
- Add 0.5 µL to each 15 µL reaction for ~1x
- Run on qPCR to determine the number of cycles to amplify… target should be the number of cycles required to reach ~1/3 of the saturation

## Step 25: Off bead amplification (library PCR 2)

- For each index, pre-mix the following
- 30 µL 2x NEB Next High Fidelity master mix
- 25 µL eluted library from Step 24
- 2.5 µL 10 uM Universal Primer (NEBNext Multiplex Oligos for Illumina)
- 2.5 µL 10 uM Indexing Primer (NEBNext Multiplex Oligos for Illumina)
- Run the following PCR cycle
  - ○ 98°C for 30 s
  - ○ 63°C for 30 s
  - ○ 72C for 40 s
  - ○ 98C for 3 min
  - ○ then N cycles of

NOTE: the number of additional cycles, N, is ascertained by estimating the number of additional cycles required to reach 1/3 saturation from the qPCR analysis in Step 24. The number of cycles will vary between each sample library.

## Step 26: High and low size selection using AMPure XP beads (final polishing)

- Allow AMPure bead slurry to warm to room temperature
- Add 0.5 volumes of AMPure bead slurry (30 µL) to PCR mix (60 µL). Mix by pipetting and incubate at RT for 5 min.
- Pull down beads with magnet, give 1–2 min to clear solution, then recover the supernatant. This should contain fragments from 0 to 500 bp. Larger fragments, if you want them, are bound to the beads. Add 70% EtOH to the beads and set aside.
- To the supernatant (90 µL), add 0.5 volumes of original volume (30 µL) for a final bead to input ratio of 1:1 (total volume should be 120 µL). Mix and incubate at RT for 5 min.

- Pull down the beads and remove the supernatant, which should contain small, unwanted fragments (adaptor dimers, etc). Nonetheless, save it for now just in case and set aside.
- Wash the beads with 700 µL 70% EtOH, pull down the beads in a magnetic tube rack until clear (1–2 min), discard supernatant, and repeat for a total of two washes. Do not disrupt the beads during the second wash.
- Remove the EtOH completely and let the beads passively dry for ~5 min
- Add 50 µL of elution buffer (20 mM Tris pH 8). Incubate for 2–3 min. Pull down the beads until clear and remove the supernatant. This should contain fragments from ~100–500 bp.

## Step 27: Quality control

- QC library by Bioanalyzer (1:10 dilution, High Sensitivity)
- Quantify library by phiX qPCR

## Sequencing

ChAR-seq libraries were sequenced with single-end 152 bp reads on the Illumina MiSeq and Next-Seq according to manufacturer's instructions.

## Data processing and software

Reads were processed using a custom pipeline, which can be accessed at: https://github.com/straightlab/flypipe (*Bell, 2017* copy archived at https://github.com/elifesciences-publications/fly-pipe). PCR duplicates were removed using Super Deduper (*Petersen et al., 2015* using the entire read length. Adapters were removed and reads trimmed for low quality with Trimmomatic (*Bolger et al., 2014*) using a composite set of Illumina adapters, leading/trailing cut length of 3, sliding window of 4:15, and a minimum length of 36. RNA and DNA portions of each read are identified and split by a custom python script that finds the bridge, uses the bridge polarity to identify which side of the read is RNA or DNA, and verifies that there is only a copy of the bridge. All reads lacking a full bridge sequence or that contained two or more bridges were removed. Only reads that contained sequence on both sides of the bridge (RNA and DNA) were processed further. Unique read IDs from the initial read containing the full molecule remained linked to both the RNA and DNA sequences after splitting from the bridge sequence. Additional software used by FlyPipe or during preparation of figures include: bowtie2 (*Langmead and Salzberg, 2012*), SAMtools (*Li et al., 2009*), BEDtools (*Quinlan and Hall, 2010*), MACS2 (*Zhang et al., 2008*), Circos (*Krzywinski et al., 2009*), R, GraphPad Prism v7.0, deepTools2 (*Ramírez et al., 2016*), HOMER (*Heinz et al., 2010*), and Genieous (*Kearse et al., 2012*).

## ChAR-seq DNA and RNA alignment

Four replicates were processed and merged for analysis in the following way. Reads were aligned to *Drosophila melanogaster* genome (dm3, release r5.57 using bowtie2 (*Langmead and Salzberg, 2012*) with the –very-sensitive option and allowing one mismatch. RNA was aligned separately to all dm3 transcriptomes from FlyBase (downloaded January 2016, versions r5.57): with the same bowtie2 parameters, but also first forcing sense strandedness as bridge orientation should preserve the RNA strand information, and then for reads that did not align in the sense direction, permitting antisense alignment. Reads that aligned with equal alignment score to more than one transcriptome were filtered based on a priority rank according to the following order: tRNA, miscRNA, ncRNA, transcript, three_prime_UTR, five_prime_UTR, exon, intron, miRNA, gene, gene_extended2000. For a given read, this rank preserves the annotation information from the top ranked transcripome. For example, a read aligning to the 'ncRNA', 'transcript', and 'exon' transcriptomes would receive ncRNA annotations, and be considered a ncRNA in further analysis. Reads aligning to tRNA and miscRNA groups were removed from further analysis. Reads lacking valid sense alignments to any transcriptome, but that aligned in the antisense orientation were ranked and filtered in the same manner. We note that for some mRNAs, some highly abundant reads emanating from the mRNA locus may potentially be expressed from small RNAs (e.g., unannotated snoRNA, tRNA, snRNA) overlapping the mRNA locus. Although the CME-W1-cl8+ cell line has the most normal karyotype of the commonly used Drosophila cell lines, the CME-W1-cl8+ reference genome has not been assembled and slight variation may exist in the positions of some reads due to potential genomic differences between CME-W1-cl8+ cells and the sequence fly strain used for the reference genome.

## ChAR-seq RNA-DNA contact identification

Previously linked RNA and DNA sequences with sense alignments had their information re-associated using the original read ID. These 1-to-1 links are deemed 'RNA-DNA contacts'. Any RNA-DNA contact that contained Ribosomal RNAs (rRNA) were removed from further analysis. Additional processing was then performed to remove mapped DNA contacts that overlapped regions of poor mappabilty using the modEncode Drosophila blacklist (*ENCODE Project Consortium, 2012*) (https://sites.google.com/site/anshulkundaje/projects/blacklists), repetitive regions downloaded from the UCSC Table Browser (https://genome.ucsc.edu/cgi-bin/hgTables) using the following selection parameters: clade: *Insect*, genome: *D. melanogaster*, assembly: *Apr. 2006 (BDGP R5/dm3)*, group: *Variation and Repeats*, track: *Repeatmasker*. Several RNAs that were re-annotated as rRNA in later *Drosophila* genome annotation versions were also removed.

## RNA coverage tracks

Individual RNA tracks were generated by extracting contact information from the BED-formatted table generated by FlyPipe (described above). Coverage was calculated using BEDtools for 200 bp windows, and the number of contacts was then normalized by dividing by the number of DpnII sites for each window throughout the genome. Coverage tracks for the metagene profile analysis in *Figure 4C* were similarly normalized, but were extracted and normalized using a sliding window method with 200 bp bins and 20 bp steps. For the correlation clustering in *Figure 4D*, we normalized the X chromosome signal to the autosomes by doubling the raw contacts on chrX and chrXHet before DpnII normalization to account for the presence of only a single X chromosome in the CME-W1-cl8+ cell line. Where indicated, coverage tracks were calculated for 2 kb bins, then converted to the log2 ratio of the contacts and DpnII frequency, which was subsequently used to calculate a z-score for each bin based on the whole genome mean and standard deviation for each RNA signal track.

## Hi-C and TAD analysis

HiC libraries ('Hi-C/Mock-ChAR') were performed as indicated in the methods (blue text indicates branch points in the protocol). Hi-C libraries were sequenced using 2 × 75 paired end reads and were aligned to blacklisted and repeatmasked genome using the 'local' and 'reorder' flags. The subsequent bam files were used to build the contact matrices using HiCExplorer (https://github.com/deeptools/HiCExplorer/blob/master/docs/index.rst). Matrices were built at 10 kb resolution. Data from GEO:GSE58821 were downloaded, re-aligned and processed using HiCExplorer in parallel. Plotting, TAD calling, and matrix correlation were all performed in HiCExplorer. Plots showing RNA-DNA and DNA-DNA contacts were also generated using HiCExplorer, where RNA-DNA matrices were also built at 10 kb resolution.

## TAD boundary analysis

Boundary regions were chosen as the 10 kb on either side around TAD edges (i.e., 20 kb total). TAD midpoints were calculated and rounded down to the nearest multiple of 10 kb, and the midpoint region was set as 10 kb to either side of the midpoint. Both boundary and midpoint region edges therefore align with 10 kb bins used for TAD calling. Regions overlapping the modENCODE blacklist or our manually curated blacklist were removed from analysis. ChAR-seq contacts were filtered to remove cis contacts, defined here as the RNA parent gene locus ±2 kb. Counts of DpnII sites, ATAC-seq insertion sites (5' end of sequenced fragments, shifted +4 bp on the +strand and −5 bp on the -strand of the reference genome) and ChAR-seq DNA contacts within regions were calculated using the coverage or map commands in bedtools (version 2.20.1). The number of base pairs in each region (midpoint or boundary) masked by our repeat masking blacklist was calculated using bedtools coverage and groupby commands. Subsequent analysis was done with R version 3.3.1. To match the distributions of masked base pairs per region between TAD midpoints and boundaries, 20 kb region masked lengths were divided into bins of 50 bp between 0 and 1000 bp, and 500 bp between 1000 bp and 20000 bp. The boundaries and midpoints were divided into these masked length bins; within each bin, boundaries and midpoints were subsampled to the minimum number of regions for the two region types (e.g., if in a masked length bin there three boundaries and five midpoints, the midpoints will be randomly subsampled to select three midpoints.) The bin matching was

verified with a Wilcoxon rank sum test of the total length of repeat masked regions in each region, with an expected non-significant p-value of 0.76. ChAR-seq DNA contact counts for each RNA type within each region were normalized by dividing the number of counts in the region by the number of DpnII sites in the region. *roX2* analysis of trans-ChAR contacts at boundary and midpoint regions was only performed for TADs on the X chromosome. All other RNA contact categories were analyzed for all TAD midpoint and boundary regions that fell outside of blacklisted regions.

## Correlation clustering and metaplots

We used deepTools (https://github.com/fidelram/deepTools) to calculate the correlation between coverage tracks using the multiBigWigSummary tool excluding the following chromosome regions: chrU, chrUextra, ChrYHet and chrM. Coverage was calculated using 100 kb bins unless indicated. The Pearson correlation coefficients were then clustered and plotted using the either the PlotCorrelation function in deepTools (snRNA clustering) or the heatplot function in R using the ward.D2 method and Euclidian distance function (modENCODE vs ChAR-seq correlation clustering). modENCODE datasets (M-values, wig-formated) for CME-W1-cl8+ were downloaded from data.modencode.org, and then re-binned into 2 kb windows, and filtered to remove modENCODE blacklist sites and repeat-regions, chrYHet, chrM, chrU and any bins lacking DpnII sites. modENCODE tracks were then transformed into z-scores by dividing the mean shifted (log2 M-values) by the genome-wide standard deviation of the bin depth. Metaplots were generated using the computeMatrix and plotProfile tools with a 2 kb window up and downstream of each region. Signal was then normalized to fold-change relative to mean of the random signal and re-ploted in GraphPad Prism.

## *roX1* and *roX2* chromosome X enrichment

The enrichment of *roX* RNAs bound to chrX was calculated by obtaining the number of ChAR-seq *roX1* or *roX2* RNA reads that contact chrX, and comparing this to the expected number on chrX if *roX1* or *roX2* reads were distributed randomly over total chromosomal sequence space. A one-tailed cumulative binomial distribution test was used to generate each p-value, which is the probability of obtaining the same or greater number of ChAR-seq *roX1* or *roX2* reads on chrX given a random genomic distribution.

## ChIRP-seq and ChAR-seq correlation analysis

ChIRP-seq binding profiles for *roX1* and *roX2* were obtained from published data (GSE53020_roX1_-merge.bw, GSE53020_roX2_merge.bw) (*Quinn et al., 2014*). The *Drosophila melanogaster* genome (dm3) was divided into equally sized windows for a range of distinct window sizes (50 bp to 1 MB). Read counts per window were obtained using BEDtools. ChIRP-seq and ChAR-seq data were filtered identically to remove modENCODE blacklist sites and repeat-regions, chrYHet, chrM, and any bins lacking DpnII sites. *roX1* and *roX2* origins were excluded due to extreme read pileups in ChIRP-seq data (*Chu et al., 2011*). Spearman's rank correlations of read-counts-per-bin across the genome between ChAR-seq and ChIRP-seq were performed in R and plots graphed and fitted with Prism.

## ChAR-seq and ChIRP-seq signal-to-noise comparison

Signal-to-noise (SNR) analysis for ChAR-seq and ChIRP-seq data was performed similar to the approach in *Figure 4B* of *Quinn et al. (2014)*, with minor modifications. We treat *roX* RNA contacts on the autosomes as 'noise', and dense chrX contacts as 'signal'. Due to the limited number of reads for individual RNAs and the insufficient resolution afforded by DpnII cut-site locations, we were unable call narrow (<1 kb) peaks for *roX1* and *roX2* ChAR-seq reads using standard methods (e.g., MACS2). Therefore, to calculate signal, we divided the genome into 2 kb bins and counted the number of *roX1* or *roX2* RNA-DNA contacts per bin for the 300 chrX bins containing the most contacts. ChAR-seq reads were normalized to DpnII cut sites within each bin. To define noise, we randomly selected 300 equal sized bins from autosomes. The mean RNA-DNA contact count per bin on chrX peaks ('signal') was divided by the mean RNA-DNA contact count per bin on autosomes ('noise') to produce the SNR value. To avoid complications due to extreme bin values or repetitive regions, we filtered for chr4, chrM, chrY, chr2Het, chr3Het, chrXHet, chrU, and modEncode blacklist regions (REF:modencode). The *roX1* and *roX2* loci on chrX were removed due to extreme values for ChIRP-seq data (*Chu et al., 2011*). For consistency, we re-calculated the SNR for ChIRP-seq *roX1* and *roX2*

data using the above approach and filtering, and obtained values similar to those published in *Quinn et al. (2014)*.

## ChAR-seq, ChIRP-seq, GRID-seq correlations

ChAR-seq and ChIRP-seq roX2 DNA contact signal was processed as in the SNR analysis. GRID-seq RNA and DNA raw reads were download from the NCBI SRA database (accessions: SRX1824449 and SRX1824449) aligned to the genome using bowtie2 (version 2.3.4.1) with the parameter –local, and minimum scoring threshold set to G,1,10. GRID-seq S2 cell Replicates 1 and 2 were merged after mapping. Uniquely mapping RNA reads with uniquely mapping DNA mates (samtools flag -q2) were preserved. RNA reads aligning to the *roX2* locus were designated as *roX2* RNAs, and the associated DNA coordinate as the *roX2* DNA-contact. For consistency with the other data sets, GRID-seq reads were filtered for repeats and small chromosome scaffolds identically to the ChAR- and ChIRP-seq filtering applied in the SNR analysis. We divided the genome into 2 kb bins and counted the number of *roX2* DNA-contacts in each bin, as above for ChAR- and ChIRP-seq. Restriction enzyme density can bias the number of reads within a given bin, so we normalized the raw GRID-seq read count by dividing the reads-per-bin by the number of AluI RE sites-per-bin (AluI cut map generated with hicup(version 0.5.2). This signal was then calculated for 20 kb bins across chrX, and Spearman's rank correlation performed pairwise among the three methods.

## RNA-seq library preparation

For total RNA-seq analysis, total RNA from 10 to 20 million CME-W1-cl8+ cells was purified using TriPure, then treated with 10 units of TURBO DNase (Life Technologies) at 37°C for 30 min according to the manufacturer's instructions. RNA was re-purified with TriPure, resuspended in DEPC-treated water, and quality checked by Bioanalyzer (Agilent). Ribosomal RNAs were depleted using the Ribo-Zero rRNA removal kit (Illumina), RNA was purified using Agencourt AMPure XP beads (Beckman Coulter), then cDNAs were generated, amplified, and indexed with the ScriptSeq v2 RNA-Seq Library Preparation Kit (Epicentre) according to manufacturer's instructions. Indexed libraries were quantified by Bioanalyzer and qPCR, pooled, and sequenced on a NextSeq 500 (Illumina).

## ATAC-seq library preparation

ATAC-seq using 250 K-500K CME-W1-cl8+ cells per reaction was performed with the Nextera DNA Library Prep Kit (Illumina, FC-121–1030). The reaction protocol was as previously described (*Buenrostro et al., 2013*), but without detergent or lysis incubation steps. Transposition reactions were performed at 37°C for 30 min shaking at 400 r.p.m. Libraries were purified with the QIAGEN MinElute Reaction Cleanup Kit and PCR amplified with barcoded primers. Amplification cycle number for each sample was monitored by qPCR to minimize PCR bias. PCR amplified libraries were purified with the QIAquick PCR Purification kit and excess primers removed by AMPure XP bead selection (Beckman Coulter). Final library concentrations were determined by qPCR using custom primers and PhiX sequence (Illumina) as a standard.

## ATAC-seq data processing

Six technical replicates were sequenced with 75 bp paired-end reads on the Illumina Next-seq. Illumina Nextera Adapters were removed using a custom Python script. Reads were aligned to the *Drosophila melanogaster* genome (dm3) using bowtie2 (version 2.2.5) with the parameter -X 2000. Duplicates were removed with Picard; mitochondrial reads or reads with bowtie2 MAPQ score <30 were removed using SAMtools. All replicates had high similarity so their alignment files were merged to increase library complexity before analysis with ChAR-seq data.

## Acknowledgements

This project was funded by a Stanford Center for Systems Biology (NIH P50 GM107615) Seed Grant to JCB, DJ, VIR and WLJ. JCB and DJ were also supported by NIH Ruth Kirchstein National Research Service Awards (F32GM116338 to JCB) and (F32GM108295 to DJ). JCB was also supported by the Stanford School of Medicine Dean's Fellowship. VIR was supported by the Walter V and Idun Berry Fellowship. NAT was supported by the Stanford Genetics Training Program (5T32HG000044-19).

OKS was supported by the Molecular Pharmacology Training Grant (NIH T32-GM113854-02). WLJ was supported by a NIH T32 Training Fellowship (GM007276) and the National Science Foundation Graduate Research Fellowship (DGE-114747). We acknowledge support from NIH RO1 HD085135 to JMS and AFS, HHMI-Simons Faculty Scholar Award to JMS, NIH grants P50HG00773501 and R21HG007726 to WJG, R01GM106005 to AFS and R01 HG009909 to WJG and AFS. We would like to thank Julia Salzman, Kyle Eagen, and members of the Straight, Greenleaf and Skotheim labs for thoughtful discussions. We thank Lucy O'Brien for sharing equipment. We acknowledge the Drosophila Genomics Resource Center (NIH grant 2P40OD010949-10A1) for providing cell lines and the Stanford Functional Genomics Facility for providing sequencing services.

## Additional information

### Funding

| Funder | Grant reference number | Author |
| --- | --- | --- |
| National Institutes of Health | Stanford Center for Systems Biology (NIH P50 GM107615) Seed Grant | Jason C Bell<br>David Jukam<br>Viviana I Risca<br>Whitney L Johnson |
| National Institutes of Health | NIH Ruth Kirchstein National Research Service Award (F32GM116338) | Jason C Bell |
| Stanford University School of Medicine | Dean's Fellowship | Jason C Bell |
| National Institutes of Health | NIH Ruth Kirchstein National Research Service Award (F32GM108295 ) | David Jukam |
| National Institutes of Health | Stanford Genetics Training Program (5T32HG000044-19) | Nicole A Teran |
| Stanford University | Walter V. and Idun Berry Fellowship | Viviana I Risca |
| Stanford University | Grant Reference: Katharine McCormick Advanced Postdoctoral Fellowship | Viviana I Risca |
| National Institutes of Health | Molecular Pharmacology Training Grant (NIH T32-GM113854-02) | Owen K Smith |
| National Institutes of Health | NIH T32 Training Fellowship (GM007276) | Whitney L Johnson |
| National Science Foundation | Graduate Research Fellowship (DGE-114747) | Whitney L Johnson |
| National Institutes of Health | RO1 HD085135 | Jan M Skotheim<br>Aaron F Straight |
| Howard Hughes Medical Institute | HHMI-Simons Faculty Scholar Award | Jan M Skotheim |
| National Institutes of Health | P50HG00773501 | William James Greenleaf |
| National Institutes of Health | R21HG007726 | William James Greenleaf |
| National Institutes of Health | R01HG009909 | Aaron F Straight<br>William James Greenleaf |
| National Institutes of Health | R01GM106005 | Aaron F Straight |

The funders had no role in study design, data collection and interpretation, or the decision to submit the work for publication.

## Author contributions
Jason C Bell, Conceptualization, Resources, Software, Formal analysis, Funding acquisition, Investigation, Methodology, Writing—original draft, Writing—review and editing; David Jukam, Viviana I Risca, Conceptualization, Resources, Software, Formal analysis, Funding acquisition, Investigation, Methodology, Writing—review and editing; Nicole A Teran, Software, Formal analysis, Investigation, Methodology, Writing—review and editing; Owen K Smith, Formal analysis, Investigation, Methodology, Writing—review and editing; Whitney L Johnson, Conceptualization, Resources, Funding acquisition, Investigation, Methodology, Writing—review and editing; Jan M Skotheim, Resources, Funding acquisition, Writing—review and editing; William James Greenleaf, Resources, Formal analysis, Funding acquisition, Methodology, Writing—review and editing; Aaron F Straight, Conceptualization, Resources, Formal analysis, Supervision, Funding acquisition, Methodology, Project administration, Writing—review and editing

## Author ORCIDs
Jason C Bell (iD) https://orcid.org/0000-0001-5480-7975
David Jukam (iD) http://orcid.org/0000-0003-4167-2754
Nicole A Teran (iD) http://orcid.org/0000-0002-9625-5010
Viviana I Risca (iD) http://orcid.org/0000-0003-2670-8704
Owen K Smith (iD) http://orcid.org/0000-0003-0880-2801
Aaron F Straight (iD) http://orcid.org/0000-0001-5885-7881

## Decision letter and Author response
Decision letter https://doi.org/10.7554/eLife.27024.036
Author response https://doi.org/10.7554/eLife.27024.037

# Additional files

## Supplementary files
• Supplementary file 1. Data table for library sequencing. Table of read count information on four libraries used for ChAR-Seq analysis, control libraries and spike in controls. Table lists raw read counts and counts after each filtering step.
DOI: https://doi.org/10.7554/eLife.27024.022

• Supplementary file 2. Table of chromatin interacting RNAs. Table of 8822 chromatin interacting RNAs. Table lists total number of contacts for each RNA, RNA length, name of the RNA, FlybaseID, normalized RNA-DNA contact frequency (contacts per kilobase million reads, CPKM), RNA-seq expression level in FPKM, and chromatin enrichment (i.e., the ratio of CPKM/FPKM).
DOI: https://doi.org/10.7554/eLife.27024.023

• Transparent reporting form
DOI: https://doi.org/10.7554/eLife.27024.024

## Major datasets
The following dataset was generated:

| Author(s) | Year | Dataset title | Dataset URL | Database, license, and accessibility information |
|---|---|---|---|---|
| Bell JC, Jukam D, Teran NA, Risca VI, Smith OK, Johnson WL, Skotheim J, Greenleaf WJ, Straight AF | 2017 | Chromatin-associated RNA sequencing (ChAR-seq) maps genome-wide RNA-to-DNA contacts | https://www.ncbi.nlm.nih.gov/geo/query/acc.cgi?acc=GSE97131 | Publicly available at the NCBI Gene Expression Omnibus (accession no: GSE97131) |

The following previously published datasets were used:

| Author(s) | Year | Dataset title | Dataset URL | Database, license, and accessibility information |
|---|---|---|---|---|
| Quinn JJ, Qu K, Chang HY | 2014 | Domain ChIRP reveals the modularity of long noncoding RNA architecture, function, and target genes | https://www.ncbi.nlm.nih.gov/geo/query/acc.cgi?acc=GSE53020 | Publicly available at the NCBI Gene Expression Omnibus (accession no: GSE53020) |
| Ramírez F, Lingg T, Toscano S, Lam KC, Georgiev P, Chung HR, Lajoie BR, de Wit E, Zhan Y, de Laat W, Dekker J, Manke T, Akhtar A | 2015 | High-Affinity Sites Form an Interaction Network to Facilitate Spreading of the MSL Complex across the X Chromosome in Drosophila | https://www.ncbi.nlm.nih.gov/geo/query/acc.cgi?acc=GSE58821 | Publicly available at the NCBI Gene Expression Omnibus (accession no: GSE58821) |
| Li X, Zhou B, Chen L, Gao L-T, Li H, Fu X-D | 2017 | GRID-seq reveals the global RNA-chromatin interactome | https://www.ncbi.nlm.nih.gov/geo/query/acc.cgi?acc=GSE82312 | Publicly available at the NCBI Gene Expression Omnibus (accession no: GSE82312) |

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
