## [Decision Letter]

Thank you for submitting your article "Chromatin-associated RNA sequencing (ChAR-seq) maps genome-wide RNA-to-DNA contacts" for consideration by *eLife*. Your article has been favorably evaluated by Kevin Struhl (Senior Editor) and three reviewers, one of whom, Job Dekker (Reviewer #1), is a member of our Board of Reviewing Editors.

The reviewers have discussed the reviews with one another and the Reviewing Editor has drafted this decision to help you prepare a revised submission.

Summary:

This is an interesting manuscript describing a new approach for mapping RNA-chromatin interactions genome-wide. It has the potential to be a major improvement over existing methods. Three patterns of RNAs linked to chromatin are identified in flies: 1) those of nascent transcript next to their genes, 2) RNAs in trans across all genome, and 3) those on X chromosome for male cells. The method could be an important approach for the study of how RNA modulates the genome and is therefore of interest to many. Several issues related to reproducibility, sensitivity and specificity need to be addressed, as outlined below.

Essential revisions:

The authors need to address the following main points:

1) Compare CharSeq performance in more detail to previously published MARGI and ChIRP-Seq methods: what are the key difference, improvements? Can you quantify this?

2) Can you add metrics describing sensitivity and specificity of the method?

3) Please clarify concerns about how many times experiments were performed and add details on statistical analysis of reproducibility of data and analyses.

4) Can you make the results available in a more user-friendly way, i.e. make the results available as a list of RNAs and loci they associate with.

5) Explore RNAs interacting with heterochromatin, rDNA etc. These are repetitive DNA sequences, but a meta-analysis of RNAs associating with defined repeats is of considerable interest and would enhance the paper.

*Reviewer #1:*

This is an interesting manuscript describing a new approach for mapping RNA-chromatin interactions genome-wide. It has the potential to be a major improvement over existing methods. I missed a more deep quantification of background, sensitivity and signal to noise ratios and how these relate to expression levels.

*Reviewer #2:*

The manuscript by Bell et al., presents a sequencing method to map all RNA-chromatin/DNA contacts in a genome: Chromatin-Associated RNA sequencing (ChAR-seq) a proximity ligation and sequencing method. The approach is utilized mainly to examine what RNAs are interacting with chromatin in *Drosophila melanogaster* CME-Wl-cl8+ (male) wing disc cells. Some work is also done using female Kc167 cells. Three patterns of RNAs linked to chromatin are identified: 1) those of nascent transcript next to their genes, 2) RNAs in trans across all genome, and 3) those on X chromosome for its inactivation. This last group, as expected, does not show up in the female cell line. The data obtained via ChAR-seq are in agreement with those obtained by Quinn et al., 2014 using ChIRP-seq. The ChAR-seq appears as a more sensitive method because it captures all RNA-DNA contacts. Authors also use their data to provide novel insights about the function of the various RNAs that they found to have abundant association with chromatin. The application of ChRA-seq in this study have revealed many snoRNA molecules interacting with chromatin in the heterochromatin form.

1) Overall, the authors show that ChRA-seq works well to identify both RNAs that are nascent and thus still connected with DNA in cis, and RNAs that do not directly interact with DNA, but rather with chromatin factors in trans. The ChRA-seq approach is validated with the RNAs that are known to be functioning in chr. X dosage compensation, in fact these RNAs (roX1 and roX2) are found to bind chr. X chromatin in the *Drosophila* male cells but not in the female cells. In addition to the similarity of the ChRA-seq approach to the ChIRP-seq by Quinn et al., Nat Biotech 2014, which although is limited to RNA-chromatin interactions, as pointed out by the Authors, ChRA-seq is almost identical to the method used in Sridhar et al., 2017in vivo. Thus, the novelty of the study by Bell et al. is in part reduced because of this recent publication, which exploits a very similar ligation method to capture sites of RNA interaction with chromatin and DNA.

Other points:

2) What is described in the first paragraph of Results does not match perfectly with Figure 1A, because the RT step in the figure is shown after the bridge is ligated to DNA, while in the text it is described before the bridge is ligated to DNA. The text should match the figure description.

3) It would be good to add the polarity of the RNA molecule in red in the Figure 1A. It would be good also to explain that the approach captures the 3' RNA ends of RNA molecules. Also Figure 1A should be bigger to clearly show which RNA end is ligated to the bridge, to clearly indicate the polarity of ends and the structure at the junction. If the App has a free 3' end, and this is ligated to 3' end RNA, is this a 3'-3' ligation? Or is the App removed in the ligation process? This should be explained well because it is not evident.

4) The statement "the 5'-adenylated end (5'App) enables increased ligation specificity for 3'-terminated ssRNA" is not clear: 'increased specificity' relative to what?

5) Is the sequence of RNA and DNA oligos presented in Figure 1—figure supplement 2 provided? It is not clear what the molar excess of DNA over RNA is, please clarify. Also the legend of this figure should explain what is shown in every lane. In lane 1 there is a high mw band, what is this, what is its size? The ladder sizes are provided only for the bottom part of the gel, they should be provided also for the top part, considering that there is a high mw band. How many times was this experiment repeated? Is there any statistical analysis of these data? This, possibly, should be explained in the legend.

6) The statement: "This RNase-treatment dramatically reduced in the number of bridge molecules identified, demonstrating that bridge ligation is indeed RNA-dependent (Figure 1—figure supplement 5)" is not strongly supported by the data shown in this figure. In addition to a typo in this statement, looking at the figure, there seems to be a factor of six reduction, which does not represent a 'dramatic' reduction. Importantly, there appears to be no repeats of this experiment and no statistical analysis of the data. Was this experiment reproducible, are there repeats that can be shown?

7) Legend to Figure 1E has a typo: "Zoomed in region of shown".

8) Typo in Results: "we performed RNA-seq to for".

9) The correlation between expression level of the RNA and FPKM contacts is not very clear, because in Figure 2—figure supplement 1 and 2 this does not seem to be the case.

*Reviewer #3:*

This manuscript presents a very important new technique (CHAR-seq) developed to identify chromatin-associated RNAs. The major benefit of this method in contrast to previously used techniques is the use of proximity dependent ligation to generate sequences from ssRNAs linked to nearby DNA, which allows relatively unbiased assessments of associations across much of the genome. The authors demonstrate very nicely that application of ChAR-seq to *Drosophila* cultured cells identifies a spectrum of chromatin-associated RNAs, including chromosome-specific, nascent mRNAs, and sn/sno RNAs. The utility and novelty of ChAR-seq makes it completely appropriate for publication in *eLife*; I am very enthusiastic about acceptance once some issues about the presentation and analyses are addressed / clarified.

1) Data completeness and transparency:

To provide a more complete resource for the community, all potentially interesting RNAs should be named/identified in a table, including those that were initially included but didn't match criteria for 'chromatin associated'.

a) There should be a table listing the 1797 RNAs (Figure 2D), and relevant information, such as% and numbers of *cis* and *trans* contacts, RNA expression level, RNA-DNA contacts, and a way to access information about their genomic distributions, etc. – essentially anything that may be of use to scientists interested in whether or not their favorite RNA is present, or in performing their own analyses of the data.

b) I assume Figure 2E shows all 1797 as 'included', but the 'chromatin associated' RNAs in Figure 2E should be listed in a table. My impression is that some (73) but not all are listed in Figure 2—figure supplement 1 – are these the red dots in 2E? Even if all red dots are listed in the supplement, readers should be able to access identity and other data for all 1797, and especially the grey dots that are RNA-DNA contact outliers with high expression (I suspect that normalization to expression levels eliminates interesting candidates, see statistics discussion below). Similarly, there appear to be RNAs with low expression that may be chromatin enriched, yet are not highlighted in red.

2) Genomics, Analysis and Statistics:a) Figure 1D, 2A and elsewhere. I could not find a description of what the black circles and darker grey areas on the chromosome graphics indicate. I assume black=centromere, but the rest is unclear.

b) Why was release 5 used, given that the more complete release 6 has been available for at least a year? In addition, genomes from the cell lines used have not been assembled, and are likely rearranged (plus other variation, including DNA copy number) relative to the reference sequence, so the positions shown in the figures are likely to be incorrect, though we don't know where. No, the authors do *not* need to assemble the clone8 or Kc genomes, but they should acknowledge this issue.

c) It is unclear what parts of the genome assembly were included, for example were heterochromatic regions included, and if so where is that data presented? This information would help with the interpretations, especially in Figure 4D.

d) A related point, for Figure 4D it would be helpful to report the genomic locations (middle of euchromatin? Pericentromeric? X vs. autosomes?) for RNAs whose chromatin contacts show strong correlation with specific chromatin marks. Also, please explain how the correlation signals were aggregated in Figure 4D – the methods described using 2kb bins while the figure showing 100kb bins.

e) Can the authors discuss how the use of PCR to amplify ligated RNA/DNA contacts, combined with normalization to expression levels, could exclude RNAs with lower transcription levels, even if they have many (potentially important) DNA contacts? I think they are being appropriately conservative, but as with all such screens, and to help others who will certainly try it themselves, it would be good to have a brief discussion about the tradeoffs.

f) Some of the statements need quantitative support, such as:

- Figure 2E – what is the threshold for identifying RNAs with more than expected chromatin interaction (how was the expectation calculated?

- How were p-values for rox1 and rox2 (7.6 fold and 8.1 fold enrichment) calculated?

- Figure 3E: correlation between the DNA contact locations from ChIRP and ChAR- seq –.is this simply the Pearson correlation coefficient?

- Figure 4—figure supplement 1: how was the base sequence similarity calculated and used to aggregate signals for different snRNAs?

g) Using the correlations between ChIRP and ChAR-seq to assess the resolution of ChAR-seq does not seem to be appropriate, since the preparation of sequencing libraries may be very different between these two datasets. (Also, the curve actually goes down as window size increases?!) The authors should provide more reasoning for this approach.

How were the replicates analyzed?.Were they merged? What are the correlations between replicates?

h) How was the z-score was calculated? It seems like it is calculated as comparing between the values in each bin vs. the rest of the genome. If this is the case, there may be no need to calculate a z-score as every window will have the same denominator, and it may be just as informative as using the read counts. Also, it may be informative to use the chromosome-specific mean (instead of whole genome mean) to further identify interesting local events (the current analysis may be mainly identifying between chromosome differences).

---

## [Author Response]

Essential revisions:The authors need to address the following main points:1) Compare CharSeq performance in more detail to previously published MARGI and ChIRP-Seq methods: what are the key difference, improvements? Can you quantify this?

We have now quantitatively compared our data to ChIRP-seq and MARGI with new analysis in the main text and figures. We have also included a comparison to another recently published method called GRID-seq (described below).

Comparison to ChIRP-seq:

ChAR-seq produces multiplexed RNA-chromatin contact data for every RNA in the cell without any requirement for the identity of the RNA, whereas ChIRP-seq requires production of oligos against a single target RNA of known sequence. They are complementary technologies, in that ChAR-seq is suited for discovery of interactions and interaction patterns in an unbiased assay, while ChIRP-seq provides targeted, higher-coverage data to explore the contact profiles of RNAs that have already been identified as interesting targets.

To address this point, we added a signal-to-noise calculation between ChIRP-seq and ChAR-seq for both roX1 and roX2 (Figure 3F). Briefly (see subsection “ChAR-seq and ChIRP-seq signal-to-noise comparison”), we used the same signal-to-noise (SNR) metric employed by Howard Chang’s Lab in a related ChIRP-seq paper (Quinn et al., 2014). This metric considers the chrX bins or peaks with the most roX RNA binding events to be ‘signal’ and treats roX RNA binding in randomly chosen autosomal bins to be ‘noise’. To quantitatively compare these two very different methods (genome-wide vs single-RNA), we were careful to re-analyze published ChIRP-seq signal from the original paper (Chu et al., 2011) with identical filtering and random bin selection. This resulted in higher SNR values for ChAR-seq than ChIRP-seq (roX1: 49.8 to 3.9, roX2: 45.2 to 22.0), despite having ~100-fold fewer roX reads per data set. We believe this quantitative analysis demonstrates the high sensitivity of our genome-wide method, in addition to validating it as identifying specifically bound RNAs.

In addition, we previously included several direct comparisons between ChIRP- and ChAR-seq, including genome-wide correlation plot for roX1 and roX2 across different bin sizes ranging over 5 orders of magnitude (Figure 3G) that revealed strong correlation over appropriate bin sizes (Spearman rho = 0.7-0.85). We also display roX1 and roX2 RNA binding signal across a 250kb region of chrX for both methods (Figure 3E).

Comparison to MARGI:

MARGI and ChAR-seq both ligate a 5-App oligonucleotide ‘bridge’ or ‘linker’ to the 3’-end of RNA. In MARGI, the linker molecule is then ligated to either restriction digested genomic DNA or sonicated DNA. Beyond this, the methods differ greatly. In MARGI, the proximal RNA and DNA are attached via circularization, and the linker molecule is cut with a restriction enzyme to form templates for sequencing primers at the end of the molecule. We list the two primary advantages of our method over MARGI in the main text discussion: first, ChAR-seq is performed in situ in intact nuclei, whereas the MARGI proximity ligations are performed on beads. As with Hi-C, it is expected that in vitro based proximity ligation will have more spurious ligation events than those from intact nuclei. Second, in ChAR-seq the final RNA-bridge-DNA molecule is sequenced with long single-end reads, which preserves the original ligation junctions and allows us to detect multiple bridges and other aberrant ligation events.

For many types of analyses, we cannot compare ChAR-seq directly to MARGI because MARGI analyzed mammalian cells, whereas we analyzed *Drosophila* cells. However, we did quantify signal-to-noise in MARGI by calculating the fold-enrichment of Xist on the X-chr as compared to autosomes:

We used MARGI data from HEK293 cells, which contain two stable Barr bodies (Gilbert et al., 2000) and thus avoid uncertainties about incomplete X inactivation associated with stem cells. Because pxMARGI (proximal) data had insufficient association reads (only counting distal and inter-chromosomal contacts), we analyzed distal RNA (diMARGI) contacts on chr X vs. autosomes. We assumed that autosomal bound Xist constitutes noise because FISH in HEK293 cells does not show widespread localization of Xist RNA (Chow et al., 2007). Because the data were also too sparse to call high-confidence X association peaks within the X chromosome, we divided the genome into 1 Mb bins, keeping only bins that (a) contained at least 1 diMARGI Xist read, (b) did not contain any ENCODE blacklisted regions, and (c) were the full 1 Mb length. The X chromosome and autosomes contain similar fractions of bins with zero Xist reads: 31% and 26%, respectively. To account for chromatin accessibility or HaeIII site bias, autosomal bins were chosen to have similar accessibility (measured by ENCODE H3K27ac ChIP) and similar HaeIII site density to the chosen bins on the X chromosome.

**Author response image 1. respfig1:** Control bins on autosomes were chosen to match bins on the X chromosome in terms of HaeIII restriction sites and accessible chromatin, here measured using the proxy of H3K27ac data because direct measurements of accessibility such as ATAC-seq or DNase-seq data were not publicly available for HEK293 cells.

Xist diMARGI contact counts in these matched bins are remarkably similar across autosomes and the X chromosome, indicating a signal to noise ratio of ~1. The ratio of the medians is 1 (2 counts per 1 Mb bin). The ratio of the means is 3.4 (X:autosome) which is largely due to a single outlier bin on the X chromosome with a count of 555, as opposed to the maximum count of 13 counts in sampled autosomal bins. This X chromosome bin contains the Xist locus and although proximal diMARGI contacts are excluded, the area around the Xist gene nevertheless contains a high density of contacts. This is not surprising, but the dramatic drop in Xist contact density from the locus of transcription to other sites on the X chromosome is not in agreement with FISH data which show Xist coating the entire Barr bodies (Chow et al., 2007). We conclude that, by this metric, the signal-to-noise in MARGI is low, and ChAR-seq may have distinct advantages as an approach to determine genome-wide RNA-DNA contacts (see Discussion).

We have attached the MARGI signal-to-noise figures for reviewers, but decided not to put this analysis in the text because we did not directly analyze mammalian data.

**Author response image 2. respfig2:** diMARGI signal to noise estimate via X vs. autosome comparison in HEK293 cells. Plots of log2(diMARGI counts per megabase) for bins spanning the X chromosome and random, accessibility- and restriction site-matched bins on autosomes (left: box pot, right: scatter plot of same data).

Comparison to GRID-seq:

While this paper was in revision, another genome-wide RNA-DNA method was published (“GRID-seq”; Li et al., 2017) that uses a methodology similar to ChAR-seq. While many details differ, GRID-seq also involves proximity ligation of a directional bridge to RNA and DNA to construct an RNA-DNA contact map. One key difference is that GRID-seq uses the restriction enzyme MMeI to cut the cDNA and gDNA fragments distal to the linker/bridge, which produces identically sized fragments. This has the advantage over ChAR-seq of being able to size select and sequence only molecules with RNA and DNA information. A critical disadvantage, however, is that reads are 19-23bp, which have a greater chance of falsely mapping to the wrong place in the genome than the 20-100bp RNA and DNA fragments obtained via ChAR-seq. We now mention the above difference in the Discussion.

The available GRID-seq data (from NCBI GEO:GSE82312) is heavily processed and some contacts were removed via a background model subtraction. We attempted to perform a signal-to-noise analysis after aligning the raw reads from NCBI SRA and filtering repeats but were unable to determine an appropriate SNR metric as with ChIRP-seq. This is because GRID-seq has lower sequencing depth for roX2 reads, and therefore the vast majority of genomic bins on the autosomes have zero reads, precluding the proper calculation. However, we analyzed the correlation of signal density between GRID and ChAR, GRID and ChIRP, or ChAR and ChIRP for roX2 reads over 20kb chrX bins. GRID and ChAR-seq match well with ChIRP (GRID-ChIRP = 0.84, ChAR-ChIRP = 0.80) and with each other (GRID-ChAR = 0.79), suggesting that – at a coarse level – ChAR-seq and GRID-seq have similar ability to detect true RNA-DNA contacts. This analysis is now in Figure 3—figure supplement 1, along with signal tracks along a 2Mb region of the genome.

False RNA-DNA contact% calculation, and comparison to MARGI and GRID-seq:

In addition to the above comparisons, we spiked in exogenous RNA species (GFP, HALO tag, and MBP) at varying levels relative to total RNA and identified their DNA contacts. This provides us with a measure of false positive contacts, which was always equal to or less than 0.5% in the condition where we spike in a contaminant up to 10% of the total RNA by mass (Figure 1—figure supplement 6). This value compares favorably to the false positive values published by MARGI (~2.2%) and GRID-seq (human cells: 3.3%, fly S2 cells: 6.9%, mESCs: 4.7%), with the caveat that a cross-species analysis was used instead of exogenous RNAs.

2) Can you add metrics describing sensitivity and specificity of the method?

As mentioned above, we have performed a spike-in experiment to address this question about specificity where we added exogenous RNA to our lysed cells in order to mimic the RNA that might be released during our fragmentation step, and which could theoretically cross-ligate to distant sites owing to non-specific binding. In this experiment, we isolated the total soluble RNA after lysis and fragmentation, added back a percentage of spike in RNA by mass at 0.1, 1 and 10%, and then proceeded with our library prep and quantified the number of false positives by alignment to the reference sequences used to generate the in vitro transcribed RNAs. Even in the condition where 10% of the total RNA was a spiked in contaminant, our false positive rate was less than 0.5%, indicating that the subsequent washes and steps after fragmentation but before ligation are sufficient to efficiently remove the contaminating RNA.

We have added signal-to-noise metrics and false-positive estimates, as described above, to the text. These reveal that ChAR-seq can detect RNAs with both high sensitivity (e.g., roX1 and roX2) and high specificity (e.g., roX1 and roX2 enrichment on chrX, and low percentage of DNA contacts by exogenous spike-in RNAs). In addition, we showed that roX1 and roX2 bind near previously identified dosage compensation peaks from ChIRP-seq data.

Other evidence for high sensitivity and specificity are presented in the following figures:

- Figure 1—figure supplement 4: 80% of ligation reads are specific to RNA.

- Figure 2E: A large subset of RNAs, including roX1 and roX2, are found enriched on chromatin after controlling for expression levels (off-diagonal RNAs in red on plot).

- Figure 3—figure supplement 2, and Figure 4C, D: When comparing female Kc167 cells to male CME-W1-cl8+ cells, roX1 and roX2 are clearly found enriched in male cells, as expected.

- Figure 3—figure supplement 1: roX enrichment on chrX.

3) Please clarify concerns about how many times experiments were performed and add details on statistical analysis of reproducibility of data and analyses.

We have now explicitly stated the number of biological and technical replicates used. Critical information relevant to each ChAR-Seq library is now presented in Supplementary file 1. For the primary data set, 4 biological replicates were performed and sequenced, along with one technical replicate sequenced at much higher depth. Additional datasets generated during revision provide three new technical replicates.

We also have added several graphs that show reproducibility of independent replicates, including a comparison between ChAR-seq replicates with respect to number of RNA-DNA contacts (Figure 1—figure supplement 5), reproducibility of chromatin enriched RNAs across 3 different replicates, at 2 different expression levels (Figure 2—figure supplement 2), and reproducibility across different cell lines (Figure 3—figure supplement 2).

Finnaly, we have included all details for statistical analysis in the main text, figure legends, or Materials and methods. These include the statistical tests, p-value, null hypothesis (where appropriate), as well as a text description of how the data inputs were generated or processed.

4) Can you make the results available in a more user-friendly way, i.e. make the results available as a list of RNAs and loci they associate with.

We have now deposited a user-friendly file into NCBI GEO (accession GSE97131) that contains all RNA-DNA contacts in BED format (RNA-chr, start, stop; and DNA-chr, start, stop). This file contains 39 additional columns with information regarding:

- RNA transcript details (length, genome coordinates, transcript coordinates, flybase ID, transcription start site, etc.).

- DNA contact details.

- RNA alignment info (mapping quality, read length, read start position, CIGAR string, etc.).

- DNA alignment information.

This text file can be easily manipulated using bedtools or shell commands to extract answers to various questions related to the dataset as a whole (what are the 100 RNAs with the most DNA contacts?) or to specific RNA or DNA locations (where does my favorite RNA bind in the genome?). We expect such a file to be valuable to biologists with or without genomics analysis experience.

In addition, we now include a supplementary table (Supplementary file 2) that lists 8822 unique RNAs, as well as: their total number of DNA contacts, RNA length, common name, Flybase ID, RNA-seq expression level, normalized RNA-DNA contact number, and chromatin enrichment (over expression level).

Finally, we will provide the code for our of analysis pipeline (from raw reads to the above user-friendly file) in a publicly accessible format for download on Gitlab (https://gitlab.com/charseq/flypipe).

5) Explore RNAs interacting with heterochromatin, rDNA etc. These are repetitive DNA sequences, but a meta-analysis of RNAs associating with defined repeats is of considerable interest and would enhance the paper.

We agree that an analysis of heterochromatin and repeats would be of great interest, and a great use of the ChAR-seq method. We attempted to analyze which RNAs were bound to repetitive DNA sequences, but found that it was difficult to make any conclusions about data since: 1) much of the repetitive portion of the *Drosophila* genome is incomplete, including some centromeric sequence; 2) given 1, the sequence space covered by a given short repeat element is unknown, making it difficult to make any meaningful conclusion regarding DNA abundance after alignment to these RNAs; 3) repeats are not easily mapped, or mapped to, with current alignment tools. For example, we removed all non-uniquely mapping reads here, but would need to set different multi-mapping threshold parameters for different repeat classes. We also attempted to analyze where RNAs that emanate from DNA repeats (such as transposons) bind; specifically, do RNAs expressed by DNA repeats bind nearby, or to other repetitive DNA loci at a greater frequency? However, this analysis was also inconclusive, primarily due to the multi-mapping issue described above.

My laboratory has a strong interest in repeat regions of the genome from our studies of vertebrate centromeres Thus, we are particularly interested in pursuing this analysis in human cells where high quality models for centromere repeats have been generated by Karen Miga, Jim Kent, David Haussler and Hunt Willard and long-read sequence information has enabled the first complete assembly of the human Y centromere. However, we feel that these experiments are outside the scope of this current manuscript.

Reviewer #2:[…] 1) Overall, the authors show that ChRA-seq works well to identify both RNAs that are nascent and thus still connected with DNA in cis, and RNAs that do not directly interact with DNA, but rather with chromatin factors in trans. The ChRA-seq approach is validated with the RNAs that are known to be functioning in chr. X dosage compensation, in fact these RNAs (roX1 and roX2) are found to bind chr. X chromatin in the Drosophila male cells but not in the female cells. In addition to the similarity of the ChRA-seq approach to the ChIRP-seq by Quinn et al., Nat Biotech 2014, which although is limited to RNA-chromatin interactions, as pointed out by the Authors, ChRA-seq is almost identical to the method used in Sridhar et al., 2017. Thus, the novelty of the study by Bell et al. is in part reduced because of this recent publication, which exploits a very similar ligation method to capture sites of RNA interaction with chromatin and DNA.Other points:2) What is described in the first paragraph of Results does not match perfectly with Figure 1A, because the RT step in the figure is shown after the bridge is ligated to DNA, while in the text it is described before the bridge is ligated to DNA. The text should match the figure description.

We thank the reviewer for pointing out this discrepancy in the figure. Figure 1A has been changed to reflect the protocol as described in the text.

3) It would be good to add the polarity of the RNA molecule in red in the Figure 1A. It would be good also to explain that the approach captures the 3' RNA ends of RNA molecules. Also Figure 1A should be bigger to clearly show which RNA end is ligated to the bridge, to clearly indicate the polarity of ends and the structure at the junction. If the App has a free 3' end, and this is ligated to 3' end RNA, is this a 3'-3' ligation? Or is the App removed in the ligation process? This should be explained well because it is not evident.

We have added a polarity indicator arrow to the illustration in Figure 1B, showing how the first-strand cDNA sequence (representing the RNA) is always 5’ of the adaptor sequence top strand (which does not contain the biotin modification) and the DNA sequence is always 3’ of the adaptor sequence top strand.

4) The statement "the 5'-adenylated end (5'App) enables increased ligation specificity for 3'-terminated ssRNA" is not clear: 'increased specificity' relative to what?

We have clarified the language in the manuscript to indicate that the ligation is performed without free ATP, making 5’ adenylation a requirement for ligating the 5’ ends. This adenylation and ligation scheme ensures that only bridge 5’ ends are ligated to RNA, preventing non-specific ligation between non-adenylated RNA 5’ ends and RNA 3’ ends.

5) Is the sequence of RNA and DNA oligos presented in Figure 1—figure supplement 2 provided? It is not clear what the molar excess of DNA over RNA is, please clarify. Also the legend of this figure should explain what is shown in every lane. In lane 1 there is a high mw band, what is this, what is its size? The ladder sizes are provided only for the bottom part of the gel, they should be provided also for the top part, considering that there is a high mw band. How many times was this experiment repeated? Is there any statistical analysis of these data? This, possibly, should be explained in the legend.

We added the missing label to Lane 3 to indicate that it represents the reaction using R55K K227Q mutant T4 Rnl2tr ligase and added additional labels for the ladder used (a combination of NEB microRNA and low range ssRNA ladders). The high molecular weight band is most likely long concatemers of the ssDNA substrate created by the Thermostable 5’App DNA/RNA ligase. It is possible that this enzyme is able to remove the 3’ block from the ssDNA oligo used in this optimization experiment, or that the block was incomplete. This enzyme is known to have higher activity in ssDNA-ssDNA ligation than the other two. We have modified the figure legend to reflect this explanation and to provide more detailed information on the enzymes used. We only use these results to make a qualitative judgment about which enzyme performed best. Because the conclusion of the experiment would not be altered by further quantification, we proceeded with the approach based on this qualitative assessment. We have also updated the legend to state that the experiment was performed once.

The DNA sequence used in Figure 1—figure supplement 2 is the universal miRNA cloning linker: CTGTAGGCACCATCAAT and the RNA sequence was that of the CENP-B box: TTTCGTTGGAAGCGGGA. These details have been added to the Figure 1—figure supplement 2 legend.

6) The statement: "This RNase-treatment dramatically reduced in the number of bridge molecules identified, demonstrating that bridge ligation is indeed RNA-dependent (Figure 1—figure supplement 5)" is not strongly supported by the data shown in this figure. In addition to a typo in this statement, looking at the figure, there seems to be a factor of six reduction, which does not represent a 'dramatic' reduction. Importantly, there appears to be no repeats of this experiment and no statistical analysis of the data. Was this experiment reproducible, are there repeats that can be shown?

We have changed the manuscript text to be more specific, indicating the fold-reduction in bridge ligation events. The experiment was performed once for this control. In response to reviewer 1, question 3, we discuss why an RNase treatment could still result in the low presence of bridge-containing sequences.

7) Legend to Figure 1E has a typo: "Zoomed in region of shown".

The figure legend has been updated to fix this typo.

8) Typo in Results: "we performed RNA-seq to for".

Thank you for pointing it out. This typo has been corrected in the manuscript.

9) The correlation between expression level of the RNA and FPKM contacts is not very clear, because in Figure 2—figure supplement 1 and 2 this does not seem to be the case.

The correlation the reviewer describes can be observed in Figure 2E, in which we plot the total RNA expression (measured by RNA-seq) vs. RNA-to-DNA contacts (measured by ChAR-seq). For most RNAs, there is a strong correlation between expression level and ChAR-seq contacts observed. The red dots represent RNAs that are enriched by at least 10-fold in the ChAR-seq dataset over the average level of contacts for a given expression level. This correlation is already normalized in the bar plot on the right side of Figure 2—figure supplement 1 and the quantity being displayed is fold-enrichment of ChAR-seq contacts over the expected contacts for a given expression level. The left bar plot is the raw expression level. What this figure illustrates is that the level of chromatin enrichment for a given RNA is not correlated with the total number of contacts we observe – enrichment is not merely a property of being above some limit of detection, it is a property of the RNA itself.

Reviewer #3:[…] 1) Data completeness and transparency:To provide a more complete resource for the community, all potentially interesting RNAs should be named/identified in a table, including those that were initially included but didn't match criteria for 'chromatin associated'.a) There should be a table listing the 1797 RNAs (Figure 2D), and relevant information, such as% and numbers of cis and trans contacts, RNA expression level, RNA-DNA contacts, and a way to access information about their genomic distributions, etc. – essentially anything that may be of use to scientists interested in whether or not their favorite RNA is present, or in performing their own analyses of the data.

We agree that the 1797 (now 1952 after additional sequencing and analysis see revised Figure 2D) most abundant transcripts are of interest. We have now generated a 39 column ‘masterfile’ that contains all RNA-DNA contacts and related information, as described in our response to the editor’s Essential revisions. These data are publicly available on NCBI GEO (GSE97131), In addition, we provide Supplementary file 1 that includes the top 8822 unique RNAs, their RNA-DNA contact number (CPKM), RNA expression level, and chromatin enrichment values, which allows any scientist to search for a particular RNA of interest.

b) I assume Figure 2E shows all 1797 as 'included', but the 'chromatin associated' RNAs in Figure 2E should be listed in a table. My impression is that some (73) but not all are listed in Figure 2—figure supplement 1 – are these the red dots in 2E? Even if all red dots are listed in the supplement, readers should be able to access identity and other data for all 1797, and especially the grey dots that are RNA-DNA contact outliers with high expression (I suspect that normalization to expression levels eliminates interesting candidates, see statistics discussion below). Similarly, there appear to be RNAs with low expression that may be chromatin enriched, yet are not highlighted in red.

The red dots in revised Figure 2E are the 138 RNAs with contacts enriched 10-fold over the expression levels, and that have more than 100 CPKM (DNA contacts). As stated above, the most abundant RNAs can be extracted from Table 1, along with all RNAs above any desired fold enrichment threshold.

2) Genomics, Analysis and Statistics:a) Figure 1D, 2A and elsewhere. I could not find a description of what the black circles and darker grey areas on the chromosome graphics indicate. I assume black=centromere, but the rest is unclear.

We have clarified these graphics in the figure legends. The cartoon chromosomes depicted in Figure 1D, 2A, 3C, and 3D are scaled visual representations of the chromosomes. The black circles are the centromeres, the light gray regions are the primary chromosome scaffolds, and the dark gray regions are the heterochromatic scaffolds.

b) Why was release 5 used, given that the more complete release 6 has been available for at least a year?

Release 5 (dm3) was used for several reasons. First and foremost, the ChIRP-seq roX1 and roX2 datasets that we used to for validation and the modENCODE data that we used for deeper analysis were all generated with the dm3 build. Many available datasets in NCBI-GEO lack either the raw read files (e.g., fastq) or lack the necessary analysis pipelines to proceed from raw reads to the available processed genome coverage files. For the processed files that are not at single bp resolution, it would be difficult to convert genome coordinates from release 5 to release 6.

Second, while major improvements on dm6 over release 5 exist in the heterochromatic scaffolds, the rest of the genome has only minor changes in dm6 compared to dm5 (http://flybase.org/static_pages/feature/previous/articles/2014_07/FB2014_04.html). Finally, coordinate conversion is generally easier going from an older build to a newer one, so for the small number of affected regions or genes, researchers for whom this is necessary can use readily available conversion tools to re-map the ChAR-seq dataset, provided in NCBI GEO.

In addition, genomes from the cell lines used have not been assembled, and are likely rearranged (plus other variation, including DNA copy number) relative to the reference sequence, so the positions shown in the figures are likely to be incorrect, though we don't know where. No, the authors do not need to assemble the clone8 or Kc genomes, but they should acknowledge this issue.

We agree that the genome assemblies are lacking for some of these cell lines and have added a sentence to the Materials and methods section to indicate that there may exist variation in the positions of some datasets due to genomic differences between cell lines. We took great care in selecting a suitable cell line for a genome wide RNA-DNA mapping method: The clone 8 cell line (CME-W1-cl8+) used for the majority of the data collection in this work has a normal and stable karyotype, and is male so we could analyze the roX genes for validation. In a recent study, 19 *Drosophila* cell lines had their DNA sequenced to determine ploidy across the genome (Lee et al., 2014). Of all 19 commonly used cell lines, the CME-W1-cl8+ line had, by far, the most optimum ploidy across the genome, which was very close to 1X throughout, whereas the commonly used S2 lines were largely 4X with variable regions up to 8X ploidy. The CME-W1-cl8+ line is also one of the modENCODE lines and maintained by the *Drosophila* Genomics Resource Center.

c) It is unclear what parts of the genome assembly were included, for example were heterochromatic regions included, and if so where is that data presented? This information would help with the interpretations, especially in Figure 4D.

We have now included which regions of the genome are included in each figure legend or extended Materials and methods, including Figure 4D. For the majority of analyses, chr2Het, chr3Het, and chrXHet were included, with exceptions noted in the figure legends or Materials and methods.

d) A related point, for Figure 4D it would be helpful to report the genomic locations (middle of euchromatin? Pericentromeric? X vs. autosomes?) for RNAs whose chromatin contacts show strong correlation with specific chromatin marks. Also, please explain how the correlation signals were aggregated in Figure 4D – the methods described using 2kb bins while the figure showing 100kb bins.

The whole reference genome was used, including the heterochromatin scaffolds; however, we filtered the regions of the build determined to have high frequency mapping artifacts (i.e., the so-called blacklisted regions) and repeats, which also produce mapping artifacts. The cluster analysis in Figure 4D is genome-wide, excluding only the chrU and chrUhet scaffolds, because those are of low quality and do not represent a real biological chromosome. The DpnII normalization (i.e., z-score calculation) of ChAR-seq tracks was performed using 2-kb windows, while the Pearson correlation coefficient was calculated pairwise between each ChAR-seq track (i.e., columns in the clustergram) and each modENCODE track (i.e., rows). The Pearson correlation coefficient was calculated using 100 kb bins because we sought to use a window size slightly larger than our estimated resolution.

e) Can the authors discuss how the use of PCR to amplify ligated RNA/DNA contacts, combined with normalization to expression levels, could exclude RNAs with lower transcription levels, even if they have many (potentially important) DNA contacts? I think they are being appropriately conservative, but as with all such screens, and to help others who will certainly try it themselves, it would be good to have a brief discussion about the tradeoffs.

It is true that using PCR to amplify our molecules during the library prep could introduce biases. However, we minimized the number of PCR cycles performed by using a side PCR to estimate the minimum number of cycles required to reach a stable exponential phase. We note that by normalizing to expression levels, RNAs with lower transcription levels but many DNA contacts may actually be more likely to be called as “chromatin-bound” after our enrichment calculations. Although our current dataset may have missed some RNAs expressed at very low levels, sequencing at greater depth with sufficient library complexity can mitigate this issue, as it does with RNA-seq.

f) Some of the statements need quantitative support, such as:- Figure 2E: what is the threshold for identifying RNAs with more than expected chromatin interaction (how was the expectation calculated?

We used two cut-offs to call RNAs chromatin enriched. First, we excluded all RNAs with fewer than 10 DNA contacts based on the cumulative frequency distribution curve as well as the fact that 10 or fewer contacts is insufficient information to make biological conclusions about each RNA. Second, we required a 10-fold enrichment of ChAR-seq contacts over expression level (FPCM). We also required more than 100 ChAR-seq contacts, which we empirically found to be a rough minimum for which to explore various properties of the RNA-DNA contact (e.g., cis/trans% , broad chromosome distribution, etc.).

- How were p-values for rox1 and rox2 (7.6 fold and 8.1 fold enrichment) calculated?

This enrichment was calculated using a one-tailed cumulative binomial test. A full description of this calculation is included in the Materials and methods.

- Figure 3E: correlation between the DNA contact locations from ChIRP and ChAR- seq – is this simply the Pearson correlation coefficient?

This is Spearman’s rank correlation, and this is now noted in the Materials and methods and new Figure 3 legend.

- Figure 4—figure supplement 1: how was the base sequence similarity calculated and used to aggregate signals for different snRNAs?

This is described in the Figure 4—figure supplement 1 legend. In short, we used GeniousR7 to calculate a pairwise distance score, then clustered the data using R.

g) Using the correlations between ChIRP and ChAR-seq to assess the resolution of ChAR-seq does not seem to be appropriate, since the preparation of sequencing libraries may be very different between these two datasets. (Also, the curve actually goes down as window size increases?!) The authors should provide more reasoning for this approach.

ChIRP, RAP, and CHART are all well established RNA hybridization and chromatin pull down methods designed to map individual chromatin associated RNAs. ChIRP, in particular, has been applied in the CME-W1-cl8+ cell line, which is why we used this specific dataset to estimate our resolution. ChIRP data have a resolution of ~1-5kb, which provides a useful lower bound, and ChIRP peaks match well with 0.5-1kb ChIP-seq peaks for the dosage compensation complex protein MSL3. The reviewer is correct that the library preparation is very different, but this is advantageous for benchmarking our method: ChIRP was established as a method for mapping chromatin interaction patterns of single RNAs, and therefore comparing the performance of the highly multiplexed ChAR-seq protocol against ChIRP-seq by selecting the same target RNA from both data sets is the most direct comparison we can make.

For both roX1 and roX2, the correlation curve increases, then plateaus. The small dip in correlation between a 250 kb bin size and a 1 Mb bin size after the plateau is likely due to the bin size being so large that differences in filtering, blacklisting, etc. between ChIRP and ChAR-seq data may cause different edge effects at ends of chromosomes or in large domains flanked by repetitive regions or heterochromatin.

How were the replicates analyzed? Were they merged? What are the correlations between replicates?

The replicates show excellent reproducibility and are now presented in Figure 1—figure supplement 5. The replicates were merged for most of the analysis, as described in the Materials and methods and the statistics included in new Supplementary file 1.

h) How was the z-score was calculated? It seems like it is calculated as comparing between the values in each bin vs. the rest of the genome. If this is the case, there may be no need to calculate a z-score as every window will have the same denominator, and it may be just as informative as using the read counts. Also, it may be informative to use the chromosome-specific mean (instead of whole genome mean) to further identify interesting local events (the current analysis may be mainly identifying between chromosome differences).

We clarified this calculation in the extended Materials and methods section. The z-score is calculated to normalize ChAR-seq mapping events relative to the DpnII frequency for a given bin size.

References:

Chow JC, Hall LL, Baldry SEL, Thorogood NP, Lawrence JB, Brown CJ. Inducible XIST-dependent X-chromosome inactivation in human somatic cells is reversible. Proceedings of the National Academy of Sciences of the United States of America. 2007;104(24):10104-10109. doi:10.1073/pnas.0610946104.

Gilbert SL, Pehrson JR, Sharp PA. XIST RNA associates with specific regions of the inactive X chromatin. J Biol Chem. 2000 Nov 24;275(47):36491-4. PubMed PMID:

ENCODE Project Consortium. An integrated encyclopedia of DNA elements in the human genome. Nature. 2012 Sep 6;489(7414):57-74. doi: 10.1038/nature11247. Updated blacklist for hg38 downloaded from: http://mitra.stanford.edu/kundaje/akundaje/release/blacklists/hg38-human/hg38.blacklist.bed.gz.